# A Roadmap for the Design, Operation and Monitoring of Renewable Energy Communities in Italy

Emanuele Cutore , Alberto Fichera and Rosaria Volpe *

Department of Electrical, Electronics and Computer Engineering, University of Catania, Via Santa Sofia 64, 95123 Catania, Italy
* Correspondence: rosaria.volpe@unict.it

**Abstract:** Renewable energy communities (RECs) aim at achieving economic, environmental, and social benefits for members and for society. This paper presents a roadmap for the design, operation, and monitoring of renewable energy communities in Italy, fundamental to guide and orient any stakeholder involved in the decision-making process of a REC. The roadmap is inspired by the Deming Cycle, also known as Plan-Do-Check-Act, which provides a framework for continuous improvement and standardization of the procedures. To demonstrate the practical application of the roadmap, a real case study is presented for Italian energy communities, making full adoption of data derived from official databases and using a real urban district as a case study. The findings of phase I in the "do" stage of the roadmap indicate that the REC could lead to a decrease in carbon emissions of roughly 38% and could support 51 to 67 families through REC's revenues, depending on the installed PV capacity. Furthermore, both physical self-consumption and virtual self-consumption schemes assist in the sustainable transition of the built environment, where consumers have a significant impact on the electrical markets. Therefore, these results validate the roadmap's effectiveness in promoting an informed design and implementation of RECs while guiding energy, social, and political decisions.

**Keywords:** stakeholders; emissions; economic and social indicators; energy management; monitoring; optimization; MILP; linearization; sustainability assessment

## 1. Introduction

The European Directive 2001/2018, "Renewable Energy Directive Recast," also known as RED II, introduced the concept of Renewable Energy Community (REC) into the energy markets, fostering the diffusion of decentralized and collaborative energy production and distribution among citizens [1]. The Directive also provides financial support for electricity production from renewable sources and self-consumption. The Directive considers RECs a way to achieve environmental, social, health, and economic benefits for the territory in which they will be constituted. Therefore, RECs serve as stimuli for the decarbonization of the energy (and, more specifically, electricity) market while providing indications for a sustainable growth of urban areas.

In Italy, the transposition process of the Directive 2018/2001 began with the Milleproroghe Decree 162/2019 [2], in particular article 42-bis, then converted with Law n. 8/2020 [3]. Other related implementing measures are contained in the resolution 727/2022/R/eel of the ARERA, the Italian Regulatory Authority for Energy, Networks and Environment [4] and in the Ministerial Decree of the 16 September 2020 of the MISE, Ministry of Enterprises, and Made in Italy [5].

Italian regulations refer to RECs as legal entities constituted by the voluntary and open participation of citizens, small/medium enterprises, and municipal authorities. To this regard and also due to the incentives from the PNRR, the National Recovery and Resilience Plan, there are increasing examples of RECs in small municipalities, where

the role played by local authorities has been decisive to start the constitution and design process of RECs [6].

Members of the REC maintain their rights as consumers (such as the right to choose the electricity distributor) and may withdraw from the community at any time. Electricity can be self-consumed within the REC members, who should mandatorily be connected to the power grid under the same primary substation. From the technical viewpoint, there is no specification on the renewable sources to be chosen or on the technological system to be installed. It is, however, remarkable how the majority of already operating RECs in the Italian territory adopts photovoltaics (PV) panels. This can be explained by the advantages deriving from an already consolidated technology, with affordable costs, supported by incentives, and easily integrated in buildings. With respect to the sizing of PV, the Legislative Decree 199/2001 [7] sets a total power of up to 1 MW that can be owned by the REC. REC members may also own PV systems before the constitution process of the community, however, in this case, the participation acceptance is set to the 30% of the total installed power of the future REC [8].

Regarding the environmental, economic, and social benefits envisaged by the Directive 2018/2001, it may be stated that environmental reduction can be estimated by evaluating the amount of electrical demand satisfied from renewable sources; economic advantages derive from the valorisation of the electricity fed into the grid and of the electricity shared within the REC. Social impacts may be attained by promoting the inclusion of families with low income and in a condition of energy poverty. In Italy, this definition refers to families that encounter difficulties in sufficiently heating their houses and paying the energy bills, estimated to hover around the 14% [9].

The concepts of energy communities, energy sharing, and peer-to-peer connections paved the way for a prolific production of research and studies among the scientific community. Regarding the adopted approach, optimization models are the most widely diffused to design energy assets within RECs [10], organized self-consumption patterns [11], and distribution networks [12]. For instance, Stentati et al. [13] proposed an optimization model to compute the battery charging/discharging policies while setting points of flexible loads and controllable generators under the Italian incentive systems. PV and energy storages size are optimized in the work of Cielo et al. [14] using a multicriteria procedure. Similarly, Cosic et al. [15] elaborated a mixed-integer linear programming (MILP) model to evaluate energy flows of RECs with distributed PV and storage systems. A dedicated MILP to model energy storage systems was used by Jasiński et al. [16]; results show that batteries only partially minimize the amount of energy drawn from the power grid when fees related to the capacity market are applied. Other production technologies, such as boilers, heat pumps, photovoltaics, and thermal storages have been taken into consideration by Zatti et al. [17]. Differently, the work of Mucha-Kuś et al. [18] investigates how different co-operative strategies, implemented within the Energy Communities, can enhance the overall economic benefits. Beyond optimization techniques, genetic algorithms [19,20], simulations [21–23], agent-based models [24], and complex network theory [25] have been explored in literature for energy sharing modelling. Apart from design issues, other authors highlight the lack of energy management strategies to address the increased complexity arising from multiple distributed energy technologies [26,27] or refer to renewable sources' integration [28], energy efficiency, or carbon reduction [29]. In this perspective, Mussadiq et al. [30] proposed a rule-based mechanism with a load-generation balancing objective. Monitoring and management techniques are also analysed by Lazdins and Mutule [31] using the existing net metering systems in Latvia. Likewise, Mutani et al. [32] presented a procedure for energy consumption and production monitoring of municipalities and companies. Ghiani et al. [33] elaborated a list of metrics for RECs' performances evaluation in Italy, while REC interactions with the power grid are discussed in [34]. Other control approaches can be found in [35], where authors presented two algorithms aimed at solving the open-loop control problem. Similarly, Bianchi et al. [36] proposed a stochastic model predictive control aiming at reaching the maximum economic benefit for RECs.

Despite the various studies proposed by the scientific community, Lowitzsch et al. [37] highlighted the need for a structured and organized framework to help stakeholders involved into the implementation process of RECs. In this same direction, Zulianello et al. [38] discussed some open questions to orient development of RECs. In light of this need, this study aims at filling this gap by proposing a structure roadmap for REC design, implementation and operation. To this scope, a comparative study conducted among the aforementioned papers and reported in Table 1 highlight the scope and novelties of this research. As emerge from the cited literature, various techniques and approaches for REC modelling, management, and control have been proposed; still, as also other literature recommends, there is the need for a structured and organized framework to orient decision-makers. Indeed, albeit detailed models remain of utmost importance for RECs' design and operation, high-level and easy-to-apply tools should be at the disposal of stakeholders. This is also the case of Italy, where red tape and difficulties arising from complex procedures and authorizations negatively impact on the RECs diffusion [39].

**Table 1.** Comparative table to highlight research gaps and novelties.

| Author(s) | Roadmap | Modelling Approach | Monitoring Phase | Technologies | | | Italian Context |
|---|---|---|---|---|---|---|---|
| | | | | PV | Batteries | Others | |
| Stentati M. et al. [13] | | MILP | | x | x | | x |
| Cielo A. et al. [14] | | MILP | | x | x | | x |
| Cosic A. et al. [15] | | MILP | | x | x | | |
| Zatti M. et al. [17] | | MILP | | x | x | x | x |
| Moncecchi M. et al. [19] | | LP | | x | x | | x |
| Moncecchi M. et al. [20] | | Genetic algorithm | | x | x | | x |
| Viti S. et al. [21] | | Simulation | | x | | | x |
| Trevisan R. et al. [22] | | Simulation | | x | x | | x |
| Mussadiq U. et al. [30] | | Grid search algorithm-based | | x | | x | |
| Lazdins R. et al. [31] | | NA | x | x | | | |
| Mutani G. et al. [32] | | NA | x | x | | | x |
| Aittahar S. et al. [35] | | Open-loop control problem | | x | x | | |
| Bianchi F.R. et al. [36] | | Stochastic model predictive control | | x | | x | x |
| This study | x | MILP | x | x | x | | x |

NA: Not applicable.

Only within a holistic action plan, the mentioned quantitative tools can effectively guide the stakeholder during the decision-making processes, thus helping in choosing the best scenario among different alternatives. As can be seen in Table 1, a properly defined roadmap is lacking in the literature, especially with reference to the Italian regulations.

Under this premise, aspects related to *how* to constitute the REC, *how* to choose members, measure *to what extent* any participation can be beneficial for the entire community from the environmental, economic, and social perspective, and *how* energy performances can be monitored remain partially open. This paper aims to address these questions, by proposing structured procedures to orient inhabitants, energy system owners, public administrations, or small/medium enterprises. Differently from the previously cited literature, this research provides guidance to these stakeholders during all stages of the REC constitution process, bypassing the bias on the design and operational phases, usually the most studied within the scientific community. Indeed, the constitution of a REC depends on several factors, starting from normative and regulatory issues to technological and operational aspects. In addition, stakeholders' decisions also impact on the constitution process of RECs. As an example, municipalities may promote the inclusion of public buildings, as well as encourage the participation of residents and small enterprises, or ensure support to families in energy poverty conditions. To this scope, a renewable energy community roadmap, clarifying goals and objectives of RECs, is here defined. The roadmap is beneficial for all members of the forthcoming REC, as it is able not only to visualize the steps around its constitution, but also to visualize the temporal occurrence of any step as well as

practical duties and expected outcomes. In addition, it fosters the communication among REC members, and is useful to evaluate REC performance in terms of deviation from the targeted objectives, here declined in environmental, economic, and social targets. As a further detail, and a novelty in the field of RECs, the presented roadmap lies its foundation in the so-called Deming Cycle, known also as Plan-Do-Check-Act (PDCA), notably diffused to elaborate strategies into organizations and small/medium to big enterprises. The PDCA cycle permits the adoption of an aware approach to orient decisions, being it constituted by well-defined processes focusing on the achievement of the overall objective of enhancing any management and operation activity.

The remainder of the paper is structured as follows. Section 2 presents the roadmap for the constitution of a new REC in Italy, following the steps of the Deming Cycle (PDCA). In the same section, the mathematical formulation of the optimization model (MILP) for the maximization of the NPV for an Italian REC is introduced and six tailored KPIs are presented. After the description of the case study and of possible operating scenarios in Section 3, the results from the optimization are discussed in Section 4. Finally, Section 5 states how the most important outcomes of this work can help in filling the identified research gaps while providing limitations and future extensions of the study.

## 2. Materials and Methods

### 2.1. Renewable Energy Community Roadmap

A roadmap can be intended as a plan of informed steps to increase awareness on how successfully run a project and define long-term strategies. The roadmap itself can be specified through milestones, i.e., particular goals to be targeted to reach the final objective. From a general perspective, the main elements of a roadmap are:

- a goal, i.e., the final objective;
- activities, i.e., actions put in place by stakeholders to achieve intermediate targets;
- milestones, i.e., tangible outputs fundamental to achieve the goal;
- timeline, i.e., the temporal occurrence of activities.

The roadmap developed in this study takes inspiration from the Plan-Do-Check-Act scheme, a management structure intensively used to control processes or products and ensure their continuous improvement. According to this, the different steps of the roadmap, including milestones and intermediate activities, should be continuously monitored and revised. This allows for a dynamic roadmap, created to continuously support REC members and stakeholders. With reference to the Italian normative, the above-mentioned steps of the roadmap have been elaborated according to Figure 1. In the roadmap, a year has been identified as the temporal horizon. In this timeframe, the different activities take place in a sequential way. It is worth noting that, although specifically built to help members of future RECs in Italy, its grounding steps can be easily applied to other European countries.

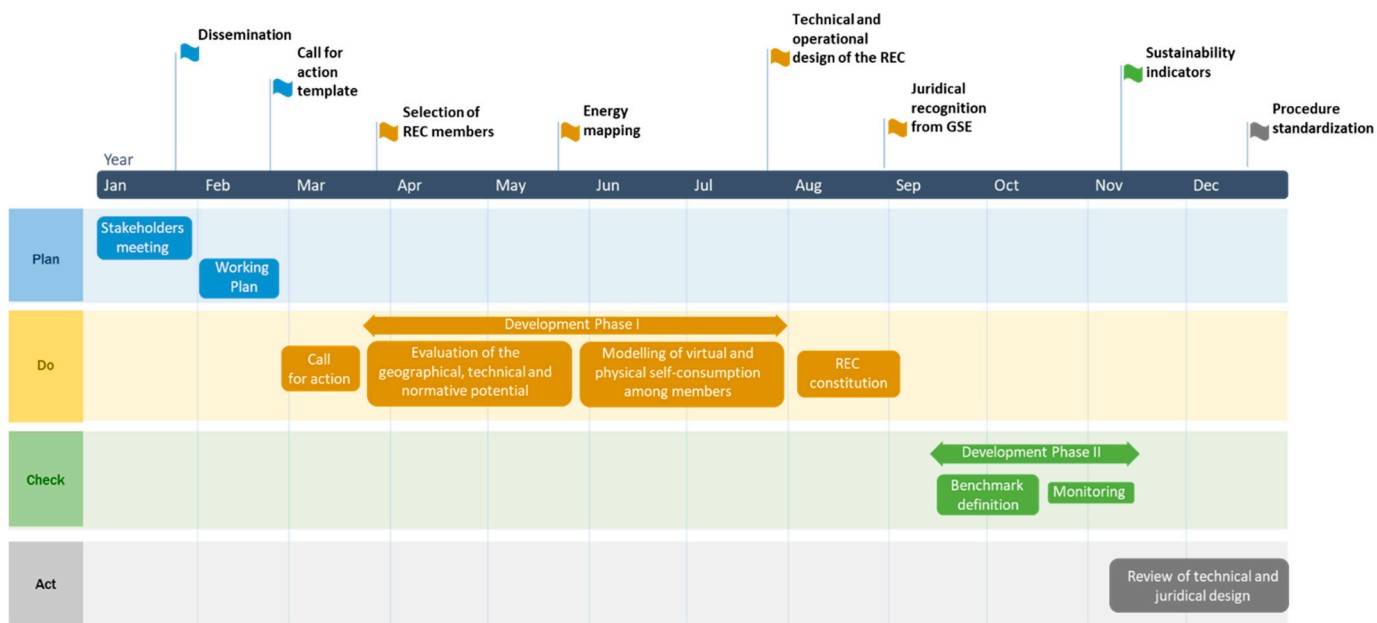

**Figure 1.** Elaborated roadmap for the constitution of RECs in Italy.

Step 1: PLAN

The first aspect to be considered is the identification of the local context in which the REC is planned to rise, starting with a study of the demographics as well as environmental, economic, and social drivers. In fact, RECs should not only target the environmental and economic conveniences, but also include the social attention to people living in slums or in a condition of energy poverty, as also suggested by the SDG7. Usually, there is always a member or a group of members taking the lead for this stage: in Italy, due also to the incentives dedicated to small/medium municipalities, it is common that local authorities or mayors seize the initiative and begin a consultation process for the engagement of citizens into the forthcoming REC. The starting point consists of the organization of dedicated meetings involving possible stakeholders, such as citizens associations, groups of consumers, retailers, and/or PV owners. The meetings have to set the basis for a common and shared vision of the REC: it is of utmost importance to gain awareness on the normative context as well as to focus on environmental concerns, expected economic revenues, and concrete actions to address energy poverty of residents. The milestone for this activity consists of a dissemination campaign, aiming at communicating to all citizens the advantages deriving from participating to a REC and raising awareness of the importance of single contributions for a sustainable transition. The subsequent step relies on the definition of a working plan to develop the strategy to be followed. The milestone for this last activity is the elaboration and writing of the call for participation that will be publicly diffused within the territory. From a temporal viewpoint, these activities may cover around two months.

Step 2: DO

Three main activities can be identified in this stage: the call for action, a development stage, and the effective REC constitution. The call for participation is made available through publication in the municipality website and through any official networks. The timeframe for participation can be arbitrarily set, but 3–4 weeks may be recommended. The publication of the call is the milestone for this activity. At the end of the call, the development phase I should (i) evaluate the geographical and technical potential of interested participants, who responded to the call, as well as verify the accomplishment of all normative aspects; and (ii) model the virtual and physical self-consumption rates among the future members of the REC. The two milestones for this activity are the energy mapping and the technical and operational design. The last activity in the "do" stage is the constitution of the REC, whose milestone consists of the juridical recognition from the GSE.

The "do" stage is the most critical for an effective and reliable design of the community and is characterized by a higher duration, usually around 6 months.

A core activity in this stage consists of the identification of the electrical loads of the participants of the community. Energy audits may be conducted to determine energy profiles and critically identify which buildings consume the most and what energy efficiency measures could be applied to save costs and energy. The audit should also focus on possible variations in future energy needs, as well as energy efficiency improvements able to reduce the energy demand. In case of difficulties in pursuing energy audits, there are methods available in the literature able to derive hourly consumption profiles for typical households from a few aggregated measures [39–41]. Data on energy demands should then be coupled with the choice of the renewable-based systems most suited for the local area in which the REC will be constituted. It is worth noting that the selection of a renewable system is not only dependent on the geographical location, it is also determined by economic and policy barriers. PV systems are likely the most suitable candidate as renewable production system for Italian cities. Indeed, from a spatial perspective, cities are usually densely populated, and rooftop areas represent a significant amount of free space, in conjunction with the high solar potential and with the attractiveness of the investment, making the PV technology sufficiently mature and competitive. Stakeholders involved in the evaluation of the solar production potential of RECs may benefit from free data portals and solar maps, such as SolarGIS [42] or PVWatts Calculator [43]. Subsequently, the technological and economic viability of the REC should be analyzed. This includes the consideration of auxiliary systems (batteries), infrastructure and equipment, costs, feed-in tariff, and subsidies. The steps explained so far belong to the operational stage of the REC constitution. However, under development phase I, it is also fundamental to determine how the geographical, technical, and normative constraints can be translated into the modelling of energy interactions at the community level. To have a better understanding of how RECs operate according to the Italian normative, Figure 2 serves as reference.

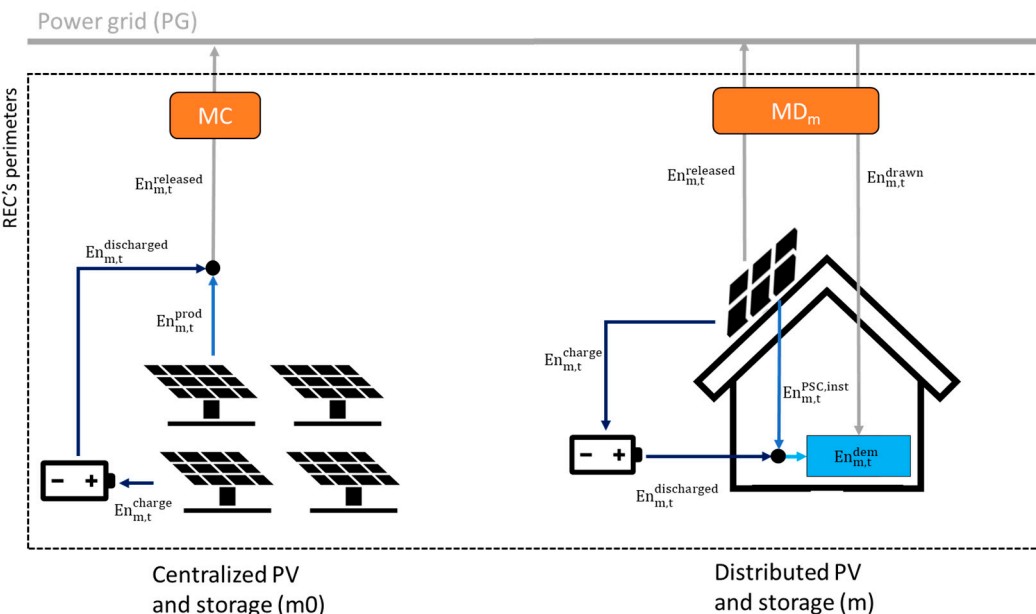

**Figure 2.** Boundaries and interactions of REC members.

According to the centralized PV and storage configuration on the left, batteries are allowed to exchange electricity only with the power grid to which they are physically connected. Since the demand associated to this centralized member is always equal to zero, the only allowed distribution direction is outgoing, i.e., electricity is only released to the power grid and never drawn from it. The amount of electricity released to the power grid is recorded on an hourly basis by a centralized meter MC owned by the REC. This

measurement will serve to compute the total electricity that is virtually shared among members by adding it to the electricity released by each member. This scheme is accounted for as virtual self-consumption, VSC, i.e., the condition for which electricity produced by the REC is virtually consumed by its members. For the distributed configuration, PVs and batteries are physically connected to the building; here the PV production is accounted for as physical self-consumed, a scheme called PSC by the normative. In addition, the PV system is connected to the power grid to release electricity in case of a surplus. In the distributed case, batteries are not allowed to exchange with the power grid. They can only be charged with electricity directly deriving from the PV discharged by electricity used by the building in which they are installed. In case of residual electrical demand, electricity is drawn from the power grid. For each distributed member, the meter $MD_m$ measures released and drawn electricity.

The mathematical model serving for the development phase I is formulated with the aim of minimizing the Net Present Value, NPV, of the investment, as expressed in Equation (1):

$$\text{maximize}\{\text{NPV}\} = -\text{INV} + \sum_y \frac{CF_y}{(1+i)^y} \tag{1}$$

The initial capital investment, INV, is equal to the sum of the capital costs for installing PVs and storages, considering the installed nominal capacity:

$$\text{INV} = \sum_m \left(\text{CAP}_m^{prod} \cdot \text{capex}^{prod} + \text{CAP}_m^{stor} \cdot \text{capex}^{stor}\right) \tag{2}$$

The yearly cash flow is given as the difference between revenues $REV_y$ and maintenance and operating costs $MOC_y$:

$$CF_y = REV_y - MOC_y, \quad \forall y \tag{3}$$

Revenues are equal to:

$$REV_y = \sum_t \left(En_t^{VSC} \cdot (val + inc) + En_t^{REC,released} \cdot sell^{PG}\right), \quad \forall y \tag{4}$$

Maintenance and operating costs can be formulated as:

$$MOC_y = \sum_t \left(\text{CAP}_m^{prod} \cdot opex^{prod} + \text{CAP}_m^{stor} \cdot opex^{stor}\right) + \sum_t \left(En_t^{REC,drawn} \cdot buy^{PG}\right), \quad \forall y \tag{5}$$

The following constraints refer to the electrical capacities and associated flows. Electrical production for each member m of the REC can be formulated as:

$$En_{m,t}^{prod} = \text{CAP}_m^{prod} \cdot conv_t, \quad \forall m, t \tag{6}$$

with the following constraint on the minimum and maximum installed nominal capacity, regulated by the binary variable $X_m^{prod}$ to control the installation of PVs for different members:

$$X_m^{prod} \cdot cap_m^{prod,min} \leq \text{CAP}_m^{prod} \leq X_m^{prod} \cdot cap_m^{prod,max}, \quad \forall m \tag{7}$$

The amount of electricity physically self-consumed, according to the PSC scheme in the Italian normative, is equal to the minimum between the electricity produced by a production technology and the electrical demand, for each member m of the REC and at each time-step t, as can be seen in Equation (8).

For future explanations, it is important to clearly distinguish the two cases that can occur:

1.  $En_{m,t}^{PSC,inst} = En_{m,t}^{dem} \rightarrow$ all demands have been covered by the PV production and an electrical energy surplus occurs.

2.  $En_{m,t}^{PSC,inst} = En_{m,t}^{prod} \rightarrow$ only a part of the total demand has been covered by the PV production and, consequently, additional electrical supply is needed to satisfy the remaining demand.

$$En_{m,t}^{PSC,inst} = \min\left\{En_{m,t}^{prod}, En_{m,t}^{dem}\right\}, \quad \forall t, m \epsilon M' \tag{8}$$

As a further specification, along with the m members of the REC, the model accounts for a "central node," labelled as $m_0$. The fictitious member $m_0$ is characterize by a nil demand, as expressed in Equation (9), and the PSC is consequently zero, as in Equation (10).

$$En_{m0,t}^{dem} = 0, \quad \forall t \tag{9}$$

$$En_{m0,t}^{PSC,inst} = 0, \quad \forall t \tag{10}$$

Equations (11) and (12) regulate the electricity balance at the single member level. These equations are strictly related to Equation (8) and two different cases may arise. In fact, in case (1), the left member of Equation (11) is equal to a positive electricity surplus that can either be used to charge the battery, $En_{m,t}^{charge}$, or released, i.e., sold, to the power grid, $En_{m,t}^{released}$. Moreover, in this case, the left member of Equation (12) is equal to zero, forcing $En_{m,t}^{disch}$ and $En_{m,t}^{drawn}$ to be zero as well. On the contrary, in case (2), the left members of Equations (11) and (12) are, respectively, equal to zero and to a positive value that represents the electricity deficit. The latter can be covered by the storage, $En_{m,t}^{disch}$, or by the power grid, $En_{m,t}^{drawn}$. These equations hold also for the central node m0. In particular, they force all produced electricity to be released to the power grid.

$$En_{m,t}^{prod} - En_{m,t}^{PSC,inst} = En_{m,t}^{charge} + En_{m,t}^{released}, \quad \forall m, t \tag{11}$$

$$En_{m,t}^{dem} - En_{m,t}^{PSC,inst} = En_{m,t}^{disch} + En_{m,t}^{drawn}, \quad \forall m, t \tag{12}$$

Moving from the single member to the community level, the total released, $En_t^{REC,rel}$, and drawn electricity, $En_t^{REC,dra}$, are calculated with the following two equations.

$$En_t^{REC,rel} = \sum_m En_{m,t}^{released}, \quad \forall t \tag{13}$$

$$En_t^{REC,dra} = \sum_{m \in M} En_{m,t}^{drawn} + En_{m0,t}^{disch}, \quad \forall t \tag{14}$$

It is worth noting that Equation (14) also includes the amount of electricity discharged by the centralized storage, eventually installed at the central node m0, to cover some excess demand of the community just before buying it from the power grid. The amount of electricity that is virtually shared under the VSC scheme at each time-step is computed as recommended by the Italian technical framework for RECs, Equation (15).

$$En_t^{VSC} = \min_t\left\{En_t^{REC,rel}, En_t^{REC,dra}\right\}, \quad \forall t \tag{15}$$

Finally, the exported and imported electricity represent the amount of electricity that is, respectively, released and drawn to and from the power grid net of the shared electricity. In particular, the latter is used to compute the actual emissions associated with the electricity production of the power grid.

$$En_t^{REC,exp} = En_t^{REC,rel} - En_t^{VSC}, \quad \forall t \tag{16}$$

$$\mathrm{En}_t^{\mathrm{REC,imp}} = \mathrm{En}_t^{\mathrm{REC,drawn}} - \mathrm{En}_t^{\mathrm{VSC}}, \quad \forall t \tag{17}$$

This final part of the model presents all the equations related to the installation and operation of storage systems. Equation (18) controls whether a storage system is installed or not at a specific member while posing a limit on its maximum capacity.

$$\mathrm{X}_m^{\mathrm{stor}} \cdot \mathrm{cap}_t^{\mathrm{stor,min}} \leq \mathrm{CAP}_m^{\mathrm{stor}} \leq \mathrm{X}_m^{\mathrm{stor}} \cdot \mathrm{cap}_t^{\mathrm{stor,max}}, \quad \forall t \tag{18}$$

The maximum charging and discharging electricity of all storage systems is limited by the energy to capacity ratio as in Equations (19) and (20).

$$\mathrm{En}_{m,t}^{\mathrm{charg}} \leq \frac{\mathrm{CAP}_m^{\mathrm{stor}}}{\mathrm{ecr}} \cdot \Delta t, \quad \forall m, t \tag{19}$$

$$\mathrm{En}_{m,t}^{\mathrm{disch}} \leq \frac{\mathrm{CAP}_m^{\mathrm{stor}}}{\mathrm{ecr}} \cdot \Delta t, \quad \forall m, t \tag{20}$$

The dynamic of the state of charge of storage systems is expressed by Equation (21), whilst Equation (22) is a time periodic constraint to plan the next period of storage. Equation (23) limits the minimum and maximum state of charge of the storage accounting for the depth of discharge, DoD, and depth of discharge, DoC, that can be found in the data sheet of each battery.

$$\mathrm{SoC}_{m,t} \leq \eta^{\mathrm{loss}} \cdot \mathrm{SoC}_{m,t-1} + \eta^{\mathrm{charg}} \cdot \mathrm{En}_{m,t}^{\mathrm{charg}} - \frac{\mathrm{En}_{m,t}^{\mathrm{disch}}}{\eta^{\mathrm{disch}}}, \quad \forall m, t \in T' \tag{21}$$

$$\mathrm{SoC}_{m,t1} = \mathrm{SoC}_{m,T} = \beta \cdot \mathrm{CAP}_m^{\mathrm{stor}}, \quad \forall m, t \tag{22}$$

$$\mathrm{CAP}_m^{\mathrm{stor}} \cdot \mathrm{DoD} \leq \mathrm{SoC}_{m,t} \leq \mathrm{CAP}_m^{\mathrm{stor}} \cdot \mathrm{DoC}, \quad \forall m, t \tag{23}$$

Finally, when a storage system is installed, the PSC of each distributed member is given by the sum of electricity coming from the PV and from the storage physically connected to the member, as expressed in Equation (24). As before, the PSC of the central node m0 is equal to zero, as expressed in Equation (25). In fact, even though electrical exchanges may occur within the community and with the power grid, no exchanges occur between the grid and any of the distributed storages installed in the REC.

$$\mathrm{En}_{m,t}^{\mathrm{PSC}} = \mathrm{En}_{m,t}^{\mathrm{PSC,inst}} + \mathrm{En}_{m,t}^{\mathrm{disch}}, \quad \forall t, m \in M' \tag{24}$$

$$\mathrm{En}_{m,t}^{\mathrm{PSC}} = \mathrm{En}_{m,t}^{\mathrm{PSC,inst}} + \mathrm{En}_{m,t}^{\mathrm{disch}}, \quad \forall t, m \in M' \tag{25}$$

*2.2. Linearization Technique*

At the moment, Equations (8) and (15) are non-linear relationships, thus adding non-convexity to the model and making it a Mixed-Integer Non-linear Programming (MINLP) problem. To linearize these equations, piece-wise linear functions can be used [44]. This can be done by introducing a new set of binary variables, $\mathrm{Y}_{m,t}^{\mathrm{prod}}$, $\mathrm{Y}_{m,t}^{\mathrm{dem}}$, $\mathrm{Y}_{m,t}^{\mathrm{released}}$, and $\mathrm{Y}_{m,t}^{\mathrm{drawn}}$, reported from Equations (26)–(30).

$$\mathrm{En}_{m,t}^{\mathrm{PSC,inst}} \leq \mathrm{En}_{m,t}^{\mathrm{prod}}, \quad \forall t, m \in M' \tag{26}$$

$$\mathrm{En}_{m,t}^{\mathrm{PSC,inst}} \leq \mathrm{En}_{m,t}^{\mathrm{dem}}, \quad \forall t, m \in M' \tag{27}$$

$$En_{m,t}^{PSC,inst} \geq En_{m,t}^{prod} - large \cdot Y_{m,t}^{prod}, \quad \forall t, m \in M' \tag{28}$$

$$En_{m,t}^{PSC,inst} \geq En_{m,t}^{dem} - large \cdot Y_{m,t}^{dem}, \quad \forall t, m \in M' \tag{29}$$

$$Y_{m,t}^{prod} + Y_{m,t}^{dem} \leq 1, \quad \forall t, m \in M' \tag{30}$$

Similarly, Equation (15) is linearized using the binary variables defined from Equations (31)–(34).

$$En_t^{VSC} \leq En_t^{REC,rel}, \quad \forall t \tag{31}$$

$$En_t^{VSC} \leq En_t^{REC,dra}, \quad \forall t \tag{32}$$

$$En_t^{VSC} \geq En_t^{REC,rel} - large \cdot Y_t^{released}, \quad \forall t \tag{33}$$

$$En_t^{VSC} \geq En_t^{REC,dra} - large \cdot Y_t^{drawn}, \quad \forall t \tag{34}$$

Step 3: CHECK

This stage consists of the development phase II, focused on the definition of benchmarks to which the REC performances should be compared during the monitoring activity. As an added-value, these tailored metrics also help for cross-comparison to other RECs, or to evaluate the impact of new members. This step usually has a duration of around 3 months. The definition of REC makes it evident that the primary goal is to attain community benefits in economic, environmental, or social aspects. Two commonly used load matching indicators are the self-consumption ratio SCR and the self-sufficiency ratio SSR, addressing technical and energetic aspects. More in detail, these indicators may evaluate how well distributed production and demand are matched in time and magnitude. SCR indicates the amount of produced electricity that is self-consumed by all members, whereas SSR measures whether the REC's needs are met by the production of PV panels, as expressed in Equations (35) and (36). SSR may also be interpreted as the degree of independence from the power grid. It is worth noting that the complementary to one SCR gives information about the amount of electricity released to the power grid, thus sold at the unit electricity price of the market. For the VSC scheme, the virtual self-sufficiency ratio VSSR is defined in Equation (37); it measures the shared electricity over the total demand. Therefore, this indicator highlights to what extent the REC production is sufficient to meet the electrical needs of members under the VSC scheme, i.e., for the amount of electricity valorized and incentivized according to the Italian normative. The indicator TSCR, called total self-consumption ratio, is built as a combination of the indicators SSR and VSSR, as reported in Equation (38). It measures to what extent the REC satisfies the electrical demand with instant PSC. These differentiations are also helpful to evaluate the economic impact for REC members in relation to the total consumption. Indeed, under the definition of the normative, PSC can be considered as an avoided expense, VSC as revenues, as it is incentivized.

$$SCR = \sum_{m,t} \frac{En_{m,t}^{PSC}}{En_{m,t}^{prod}} \tag{35}$$

$$SSR = \sum_{m,t} \frac{En_{m,t}^{PSC}}{En_{m,t}^{dem}} \tag{36}$$

$$VSSR = \sum_{m,t} \frac{En_t^{VSC}}{En_{m,t}^{dem}}, \quad \forall m, t \tag{37}$$

$$\text{TSCR} = \sum_{m,t} \frac{\text{En}_t^{\text{VSC}} + \text{En}_{m,t}^{\text{PSC}}}{\text{En}_{m,t}^{\text{dem}}}, \quad \forall m, t \tag{38}$$

Environmental performances are evaluated considering the carbon dioxide emission index, $CO_{2,\text{avoided}}$, calculated as the percentage of $CO_2$ emissions avoided thanks to the constitution of the REC, as expressed in Equation (39). In fact, when a REC is constituted, the electricity produced by renewable technologies and discharged by batteries reduces the amount of electricity imported from the grid, $\text{En}_t^{\text{imp}}$. In this paper, a net-balancing approach to carbon dioxide emissions is chosen, and, therefore, electricity sharing under the VSC scheme is considered carbon-neutral. A national-specific standard emission factor efg is associated with the electricity imported from the grid and used to calculate grid-related emissions, as in Equation (40).

$$CO_{2,\text{avoided}} = \frac{CO_2^{noREC} - CO_2^{REC}}{CO_2^{noREC}} \tag{39}$$

$$CO_2 = \sum_t \left( \text{En}_t^{\text{imp}} \cdot \text{efg} \right), \forall t \tag{40}$$

To account for the social impact of the REC, the energy poverty help index EPHI is calculated to account for the number of families in energy poverty conditions that can be financially helped thanks to the distribution of REC revenues. Therefore, the EPHI indicator can be calculated as in Equation (41). The value of AED has been fixed equal to 2700 $kWh_e/y$, that is the reference for a typical Italian family according to the Italian National Institute of Statistics [45].

$$\text{EPHI} = \frac{\text{NPV}^{\text{REC}} - \text{NPV}^{\text{noREC}}}{\left( \text{AED} \cdot \text{buy}^{\text{PG}} \right) \cdot Y} \tag{41}$$

Step 4: ACT

The fourth step of the roadmap consists of a review of the technical and juridical design of RECs. During this activity, actions may be taken with respect to the results obtained from the previous steps and, generally, from the experiences gained during the overall constitution process. Indeed, it is important to review the core activities of the other steps and identify eventual "bottlenecks" or criticisms to adjust the strategy trajectory and incorporate any lessons learned. In case any change is going to be adopted (such as new documents, model adjustments, or the introduction of new indicators), the PDCA cycle should start again and be revised according to the needs. A typical milestone of this step is the standardization of procedures, fundamental to guarantee the continuous improvement and operation of the REC and also to serve as example for other forthcoming RECs. The timeframe for this step is around 1 to 2 months.

## 3. Case Study

The steps of the REC roadmap have been applied to a real urban area of 17 residential (multiapartment) buildings in Catania, Sicily, once verified that all candidate members respect the criteria established by the Italian regulation [8]. For each building, Table 2 reports the data that have been obtained through a collection campaign carried out by the authors in the year 2022. After the building Id, the columns show in order: the available south-oriented roof surface, the maximum PV and storage capacities, and the annual electricity consumption for each building. Values in the third and fourth columns are used as input data of the optimization to define the upper bounds on the total nominal capacity of PVs and batteries that can be installed for each building, see Equations (7) and (18). The annual electricity consumption, in the last column, represents the total electrical energy demand derived from the collection campaign.

**Table 2.** Electrical demands and maximum capacities for PV and battery systems of each participant of the REC for the year 2022.

| Bulding_Id | Facing-South Surface Area [m$^2$] | Maximum PV Capacity [kW$_n$] | Maximum Battery Capacity [kWh] | Annual Electricity Consumption [kWh] |
|---|---|---|---|---|
| m0 | - | 50 | - | - |
| m1 | 31.00 | 3.00 | 2 | 1139.71 |
| m2 | 32.00 | 3.00 | 2 | 1728.25 |
| m3 | 28.00 | 3.00 | 2 | 1298.30 |
| m4 | 31.00 | 3.00 | 2 | 2548.63 |
| m5 | 30.00 | 3.00 | 2 | 3419.14 |
| m6 | 28.00 | 3.00 | 2 | 1367.65 |
| m7 | 22.00 | 2.00 | 1 | 26,952.03 |
| m8 | 23.00 | 2.00 | 1 | 26,952.03 |
| m9 | 20.00 | 2.00 | 1 | 27,032.58 |
| m10 | 21.00 | 2.00 | 1 | 27,032.58 |
| m11 | 22.00 | 2.00 | 1 | 26,935.92 |
| m12 | 20.00 | 2.00 | 1 | 26,935.92 |
| m13 | 11.00 | 1.00 | 1 | 26,919.81 |
| m14 | 9.00 | 1.00 | 1 | 26,919.81 |
| m15 | 240.00 | 24.00 | 16 | 35,046.24 |
| m16 | 238.00 | 24.00 | 16 | 35,046.24 |
| m17 | 332.00 | 33.00 | 22 | 24,461.42 |
| Total | 1138 | 113 | 74 | 321,736.3 |

The maximum installable PV capacity, reported in the third column, is related to the total available rooftop area of each building. In fact, it has been calculated by multiplying the available area, in the second column, by a typical factor of 10 m$^2$/kW$_n$ that represents the occupied surface of a PV panel of 1 kW of nominal capacity, as done in Ref. [46]. Along with the buildings of the future REC, the model includes a dedicated area suitable for PV installation and indicated with the label m0. Therefore, it has null electrical demand. The PV installation of this site is accounted to be owned by the future REC, and the balances between production and electricity consumed by the REC or sold to the power grid are managed according to the VSC scheme on an hourly basis. The maximum capacity of the battery, reported in the fourth column, is calculated considering the PV maximum capacity.

The same load scaling technique used in the research of Ref. [40] has been applied in this study to derive the hourly electrical demand profile starting from aggregated annual electricity demands and associating a typical daily electrical load curve for residential buildings, plotted in Figure 3.

Two typical electrical demand profiles derived from the load scaling technique in the first day of the year have been reported in Figures 4 and 5, for building m1 and building m15, the less and most energy intensive of the future REC, respectively.

The hourly electrical production from PV panels has been estimated using PVWatts Calculator [43]. Figure 5 reports the PV production per kW$_n$ of installed capacity for a spring, summer, autumn, and winter typical days in the Nesima district (Catania). The optimization model derives the total hourly electrical production of each building during the entire year by multiplying the values of the PV production per kW$_n$ of installed capacity and the variable CAP$_n^{pv}$, i.e., the nominal capacity of PV panels installed on buildings' rooftop areas.

The technical and economic parameters used to run the model are listed in Table 3. It is worth noting that the unit cost of electricity purchased from the power grid as well as the unit price of selling electricity to the energy market, as defined from the GSE [47], are here not included, but are used to perform the following scenario analysis.

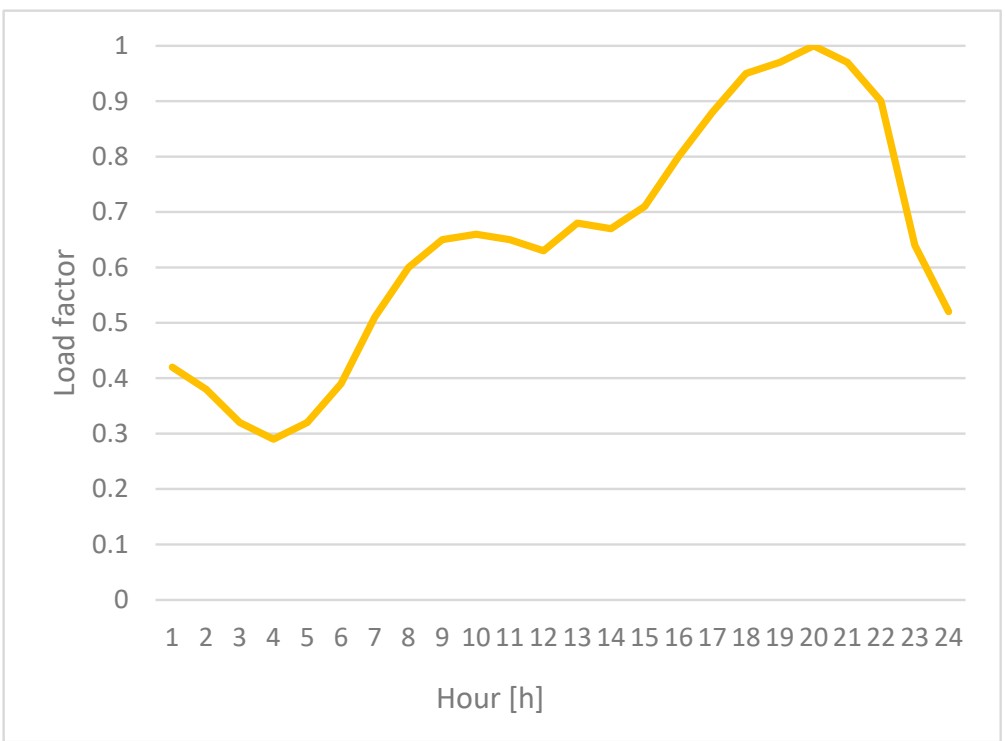

**Figure 3.** Typical daily load curve for a residential apartment.

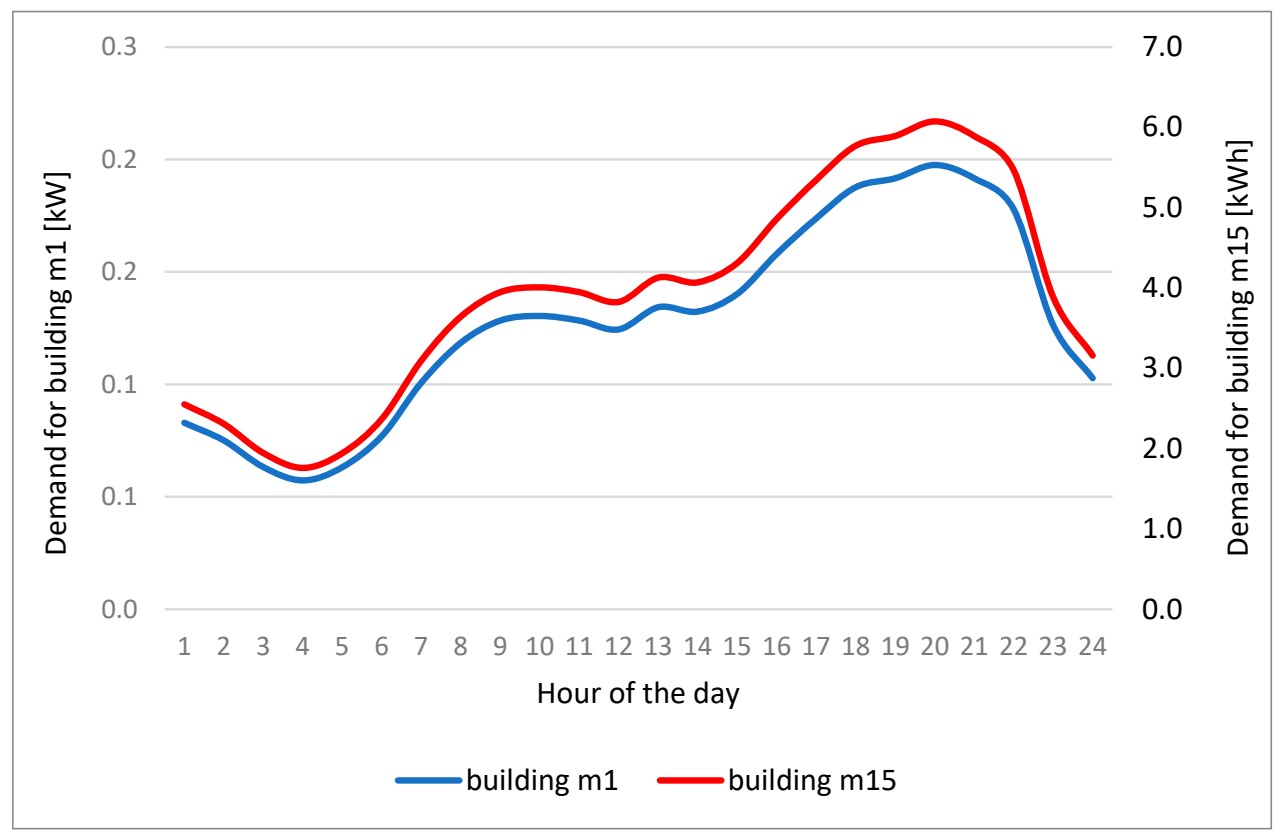

**Figure 4.** Electrical demand profile of building m1 and m15.

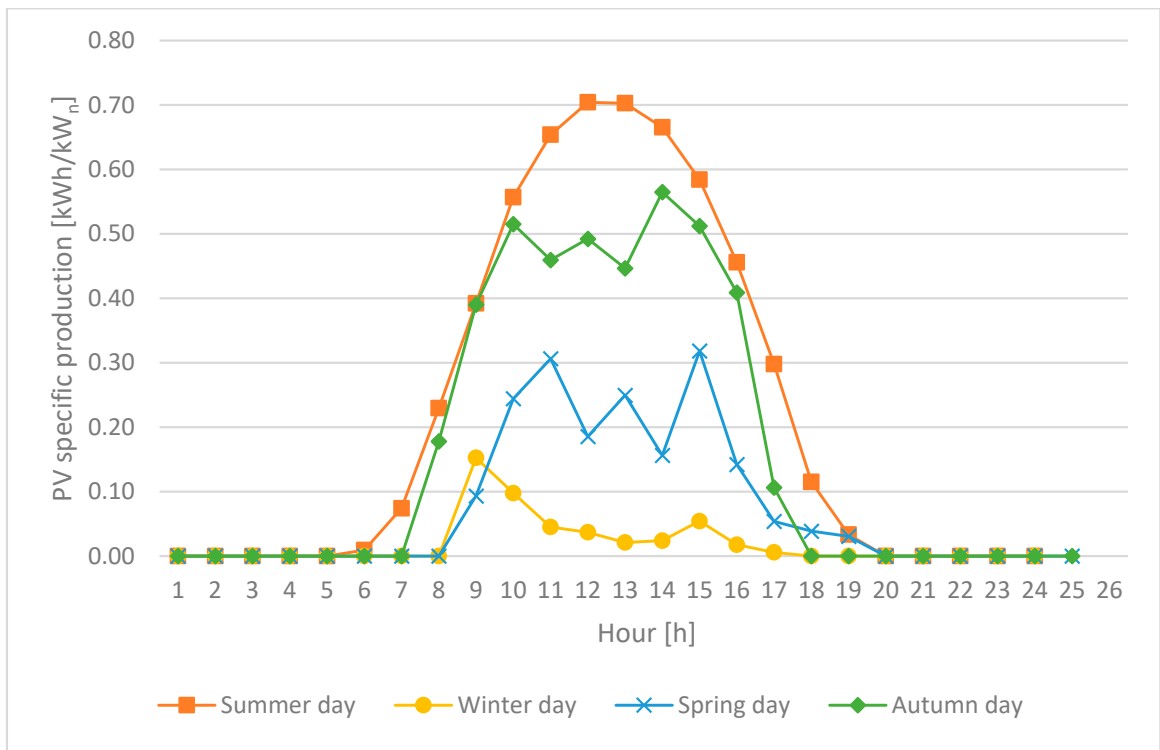

**Figure 5.** Hourly specific production profile per installed $kW_n$ of PV for four typical days (summer day, 21 June 2022; winter day, 21 December 2022; spring day, 20 March 2022; autumn day, 23 September 2022) in the Nesima district (Catania).

**Table 3.** Technical and economic parameters.

| Description (Symbol) | Value | UoM | Ref. |
|---|---|---|---|
| PV capex capex$^{prod}$ | 1200 | €/$kW_n$ | [48] |
| PV opex opex$^{prod}$ | 2% capex | (€/$kW_n$)/yr | [19] |
| PV efficiency | 14% | - | [43] |
| Battery capex capex$^{stor}$ | 1000 | €/kWh | [48] |
| Battery opex opex$^{stor}$ | 1% capex | (€/kWh)/yr | [19] |
| Depth of Discharge (DoD) | 10% | - | [49] |
| Depth of Charge (DoC) | 100% | - | [15] |
| Charging efficiency η$^{charg}$ | 95% | - | [15] |
| Discharging efficiency η$^{discharg}$ | 95% | - | [15] |
| Battery loss factor η$^{loss}$ | 2% | - | [15] |
| Energy to capacity ratio (ecr) | 34% | - | [15] |
| Valorization of shared electricity (val) | 0.00848 | €/kWh | [8] |
| Incentivization of shared electricity (inc) | 0.110 | €/kWh | [8] |
| Emission factor power grid (efg) | 0.247 | $kgCO_2$/kWh | [50] |
| Discount rate (i) | 4% | - | [19] |
| Investment life (y) | 20 | yr | [8] |
| Space occupation of PV | 10 | $m^2$/$kW_n$ | [49] |

The optimization model is used to run different scenarios, identified in Figure 6, all with the same objective function, i.e., the maximization of the NPV. Two main scenarios have been chosen: a first scenario #1, is optimized using the input data listed in Tables 2 and 3 and a second scenario #2 with an increased installed capacity on the centralized node m0, from 50 $kW_n$ to 100 $kW_n$. The node m0 has been chosen as the sole site in which the PV installation may be varied; indeed, buildings' rooftops are already limited by their area and PV capacity cannot be varied arbitrarily. Under the base scenario #1, four different

variations in terms of both unit cost of purchasing electricity from the power grid and unit price of selling electricity to the power grid have been selected. A minimum and maximum variation of 25% has been accounted for, as can be observed from Figure 6. The scenarios are labelled as #1.1 (+25% increment of unit cost of purchasing electricity from the power grid), #1.2 (−25% decrement of unit cost of purchasing electricity from the power grid), #1.3 (+25% increment of unit price of selling electricity to the power grid), and #1.4 (−25% decrement of unit price of selling electricity to the power grid). This choice is useful to evaluate to what extent the unit cost and unit price impact on the REC performances and on the NPV, which is highly influenced by these variables.

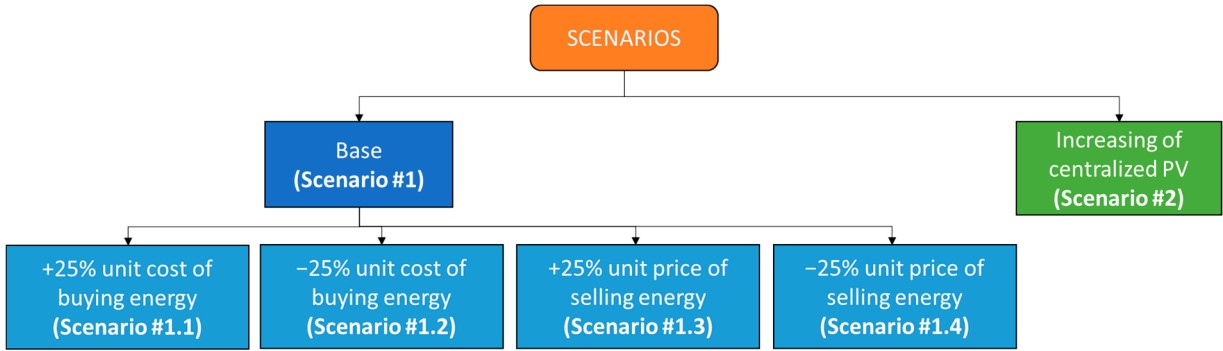

**Figure 6.** Scenarios used for optimization.

The unit cost of buying electricity from the power grid is 0.53 €/kWh [51] and the unit price of selling electricity to the energy market is 0.435 €/kWh [51].

## 4. Results and Discussion

The results of the optimized scenarios described in Figure 6 are presented and discussed here. As can be seen from Table 3, the total electrical demand of the REC is equal to 321,714.5 kWh/y. Without the constitution of the REC, the total cost for purchasing electricity from the power grid would have been €170,520 (considering the unit cost of 0.53 €/kWh [51]). In addition to this, if considering the investment for PV panels for all buildings, the NPV over a temporal horizon of 20 years would be −€2,317,422.25. These considerations are useful to discuss the economic impact of the REC. The optimized capacities of PV and storages are summarized in Table 4, for both scenarios #1 and #2.

**Table 4.** Optimized values for PV and storage installations for scenarios #1 and #2.

| Data | Value | | UoM |
|---|---|---|---|
| NPV without REC | −2,317,422.40 | | € |
| | **Scenario #1** | **Scenario #2** | |
| NPV with REC | −862,407.24 | −399,208.23 | € |
| Initial investment PV | 195,600 | 255,600 | € |
| Initial investment BESS | 0 | 0 | € |
| Annual cost | 135,515.22 | 135,514.50 | €/year |
| Annual revenues | 90,695.50 | 125,863.7 | €/year |
| Grid-related emissions $CO_2$ | 49,093.2 | 47,824.5 | $kgCO_2$/year |
| Self-consumption ratio (SCR) | 27.16% | 19.51% | - |
| Self-sufficiency ratio (SSR) | 20.52% | 20.52% | - |
| Energy shared ratio (VSSR) | 17.28% | 19.29% | - |
| Total self-consumption (TSCR) | 37.81% | 39.82% | - |
| $CO_{2, avoided}$ | 38.2% | 39.9% | - |
| Energy poverty help (EPHI) | 51 | 67 | #families |

The NPV is negative for both scenarios, still it is higher than the prospective value of the NPV without REC, as discussed above, i.e., in the case of purchasing all the electricity

from the power grid to satisfy the demands of the buildings, without the chance of self-consumption or sharing. In these terms, the advantage of constituting a REC is evident. Moreover, the NPV of scenario #2 is higher than scenario #1, as the installed PV capacity on building m0 positively impacts on the cash flow despite the higher initial investment. As can be observed, no batteries have been installed. Indeed, under the Italian regulation, the installation of batteries would not have been advantageous in terms of NPV maximization, considering that the electricity would be stored instead of sold to the power grid at a price that would be in any case higher than the price at which electricity would be shared among members [4]. Indeed, with battery installation, the shared electricity increases and, consequently, the sold electricity to the power grid decreases. As a further aspect, the initial investment of batteries should also be considered in the overall maximization of the NPV, therefore the overall income significantly diminishes in the case of batteries. Regarding the installed PV capacity, it can be observed that it corresponds to the maximum permitted values, thus confirming that, under the VSC, PV installation is convenient, despite the still significant initial investment. The $CO_2$ emissions have been calculated by associating the net imported electricity from the power grid, $En_t^{imp}$, and the national emission factor of 0.247 $kgCO_2/kWh$ [51]. As expected, some indicators do not vary when moving from scenario #1 to scenario #2, i.e., SSR and $CO_{2,avoided}$. In fact, the increase of centralized PV capacity does not affect the total PSC and all the electricity produced from the central node is released to the power grid and affects the sole VSC. As expected, the self-consumption ratio, SCR, remains unchanged moving from scenario #1 to scenario #2. Increasing the centralized PV capacity has no impact on the total PSC, as the electricity produced from m0 released to the power grid and only affects VSC, as confirmed by the increase of the indicator VSSR. In addition to that, the $CO_{2,avoided}$ increases, as it associated with imported electricity from the grid that changes with the amount of electricity shared, as in Equation (17). The NPV is positively affected as the increased electricity production is sold to the grid and also shared among REC members. On the contrary, increasing the PV capacity at the central node decreases the indicator SCR, therefore showing a minor contribution in terms of PSC. The SCR can also be discussed in relation to VSSR, reflecting the shared electricity over the total demand. Comparing these two indicators is equivalent to comparing the two main contributors to NPV: electricity savings from PSC, in the case of SCR, and incentives for shared electricity, in the case of VSSR. Of course, the total self-consumption rate TSCR, as the sum of SCR and VSSR, represents the total amount of electricity consumed under both physical and virtual schemes, thus serving as an overall performance metric of the REC, useful for monitoring, benchmarking, and more importantly, evaluating the efficacy of actions taken with the goal of continuously improving the system. Finally, the social index of energy poverty and household income EPHI increases when transitioning from scenario #1 to scenario #2, indicating that a higher economic contribution can be provided to families suffering from energy poverty within the REC's area of influence.

Going into more detail about the electricity interactions within the REC and between the REC and the power grid, it is necessary to highlight the daily exchanges under scenario #1; indeed, the variation from scenario #2 is only attributed to the VSC and, in any case, comparable with the following results. In particular, the electricity fluxes are shown at the community level and for single buildings. Starting with the community level, Figure 7 reports the daily electricity production that is released to the power grid.

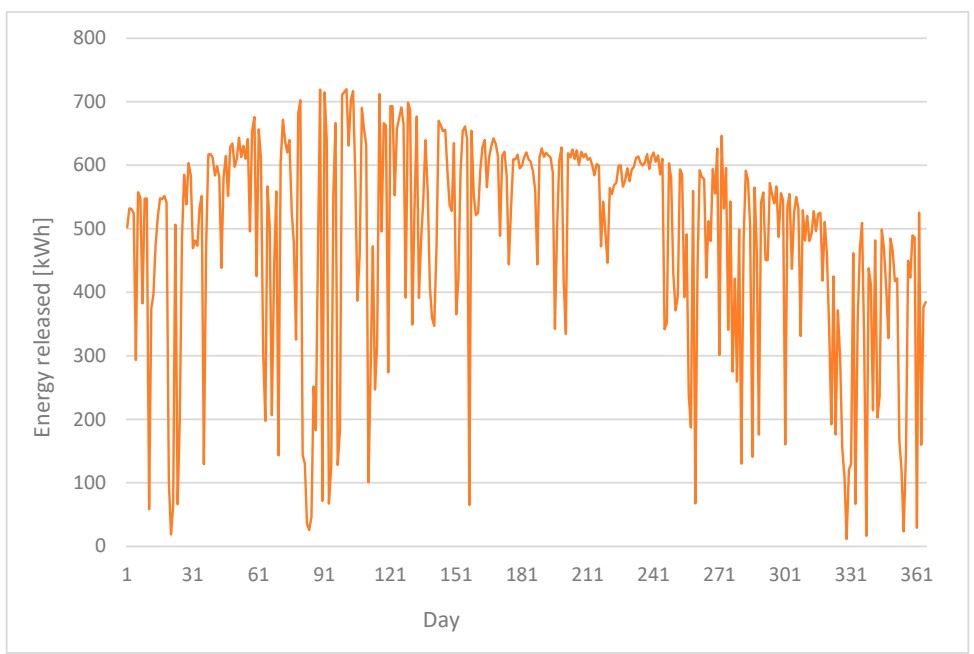

**Figure 7.** Daily electricity production released to the power grid in 2022 and scenario #1.

The exchanges at the community level have to be considered as net to PSC. In other words, during each day, the electricity exchanges accounted for in Figure 7 only regards the physical self-consumption and the electricity shared within the REC is accounted for as in the VSC scheme. As can be observed, a higher amount of electricity production released to the grid can be noticed during the spring and summer seasons, due to the higher production from PV panels. On the contrary, during winter days, the minor PV production coupled with the PSC from members yields a lower net amount of electricity that is released to the power grid. As a comparison, Figure 8 plots the daily electricity that is drawn from the power grid at the community level.

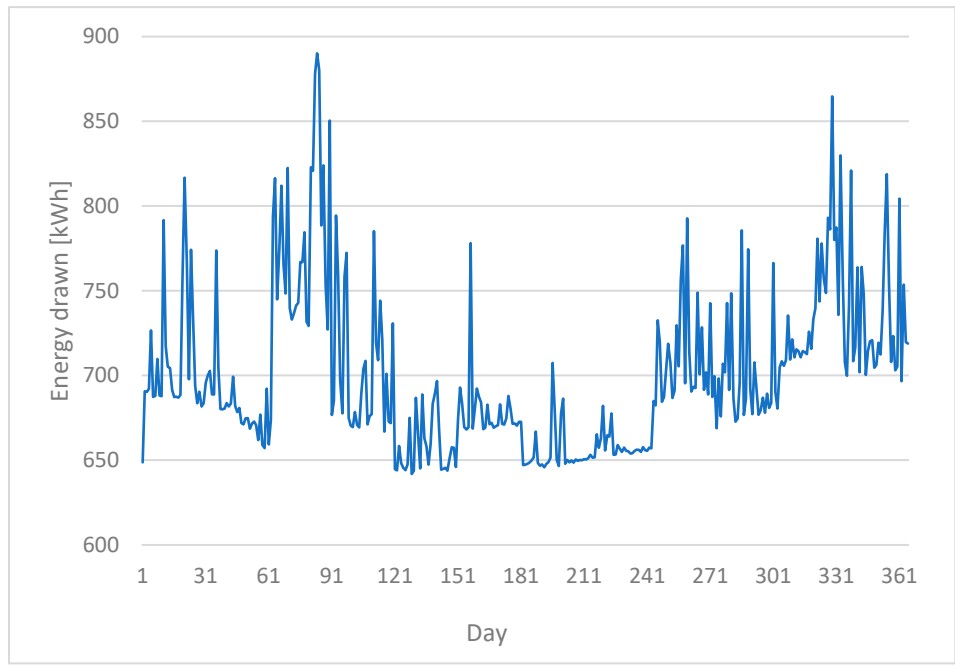

**Figure 8.** Daily electricity drawn from the power grid in 2022 and scenario #1.

It corresponds to the electricity required to fulfill the demands of the REC members that have not been satisfied with the PV production. At the same time, it is also important to discuss the electricity shared within the REC under the VSC scheme, reported in Figure 9.

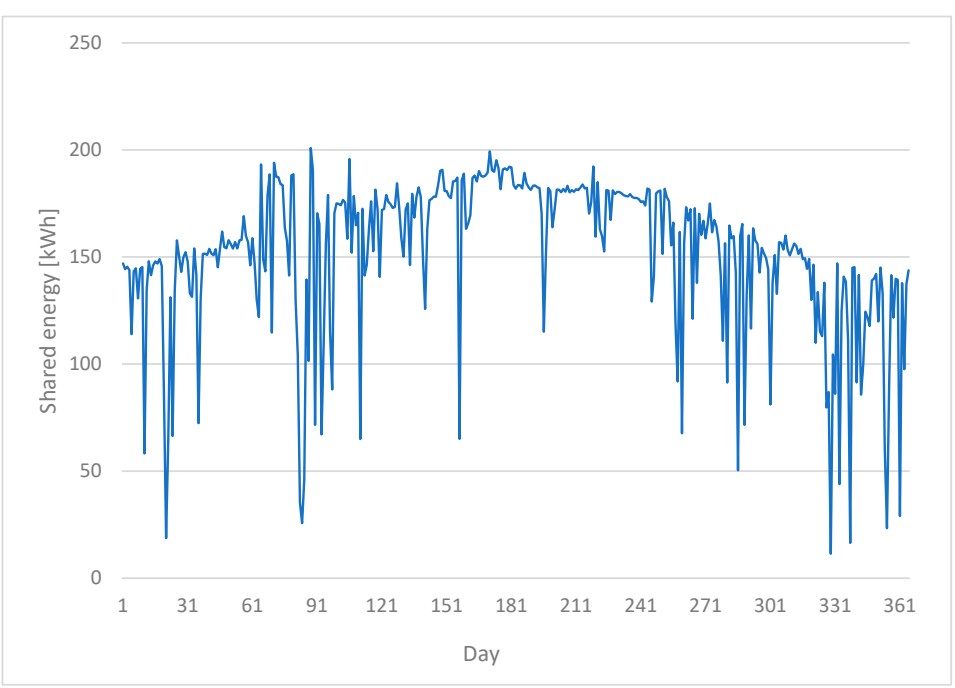

**Figure 9.** Daily electricity virtually shared within the REC in 2022 and scenario #1.

The electricity exchanges plotted in Figure 9 correspond to Equation (15) and constitute the interactions occurring within the REC. The electricity that is produced by PV panels and is not consumed under the PSC scheme constitutes the electricity released to the grid, as said when Figure 8 was commented on. The electricity released to the grid, however, also includes the contributions of the VSC, which corresponds to the electricity shared among members of the REC. As can be observed, the amount of shared electricity is higher during summer, when the production from PV is higher. In addition to this, it is worth noting that the electricity fluxes plotted in Figure 9 are incentivized from the normative, as they are identified as VSC within the REC. Therefore, the higher this amount of electricity shared among members is, the better in terms of revenues for the community.

Figure 10 plots the electricity imported and exported from the REC during the entire year.

These fluxes should be read as the electricity exported and net imported electricity shared among members. In this regard, the electricity exported to the power grid is paid but not valorized under the VSC scheme, as it not shared among members. The imported electricity, on the other side, is also responsible for the $CO_2$ emissions that are associated with the REC.

Finally, to properly account for a correct design of the electricity fluxes useful to conclude the development phase I of the "do" stage of the roadmap, it is crucial to evaluate the exchanges at the level of single buildings. For this scope, the less and most energy intensive buildings were chosen, i.e., building m1 and building m15, whose optimized results are reported in Figures 11 and 12, respectively.

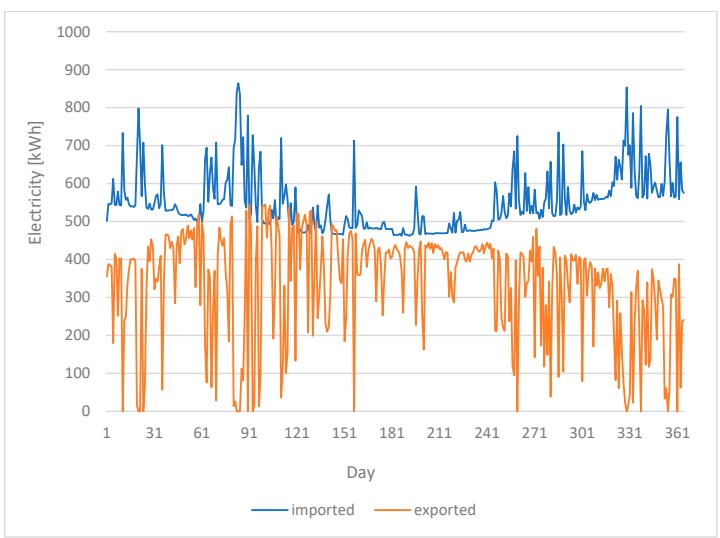

**Figure 10.** Daily exported and imported electricity in 2022 and scenario #1.

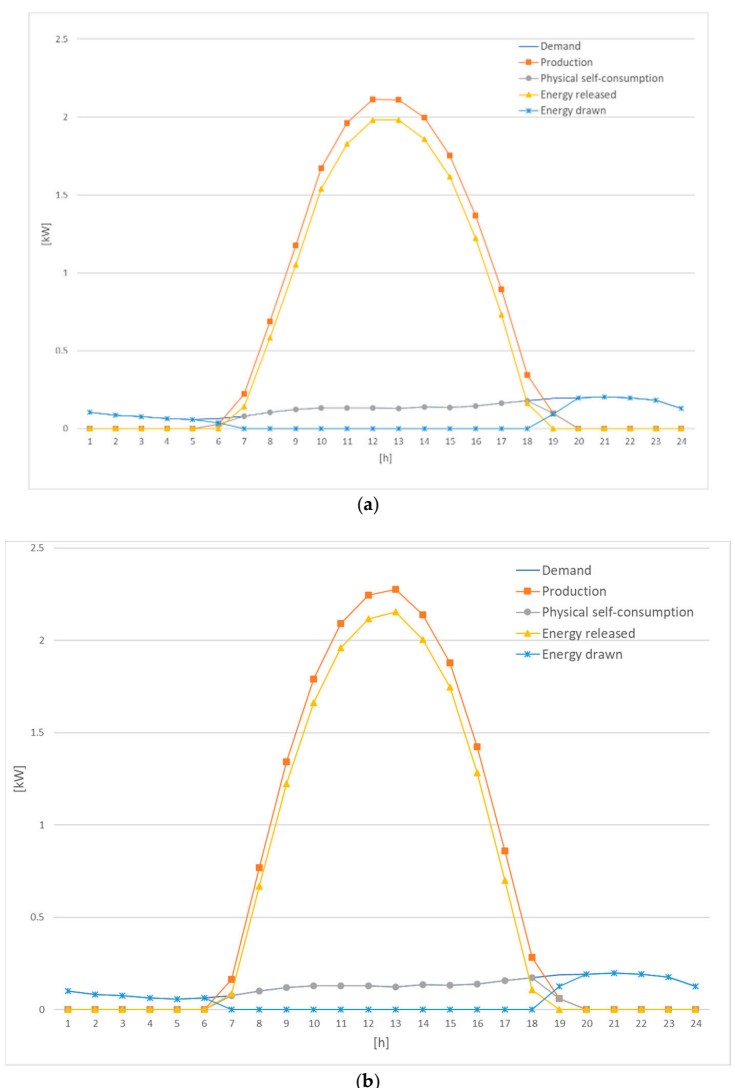

**Figure 11.** *Cont.*

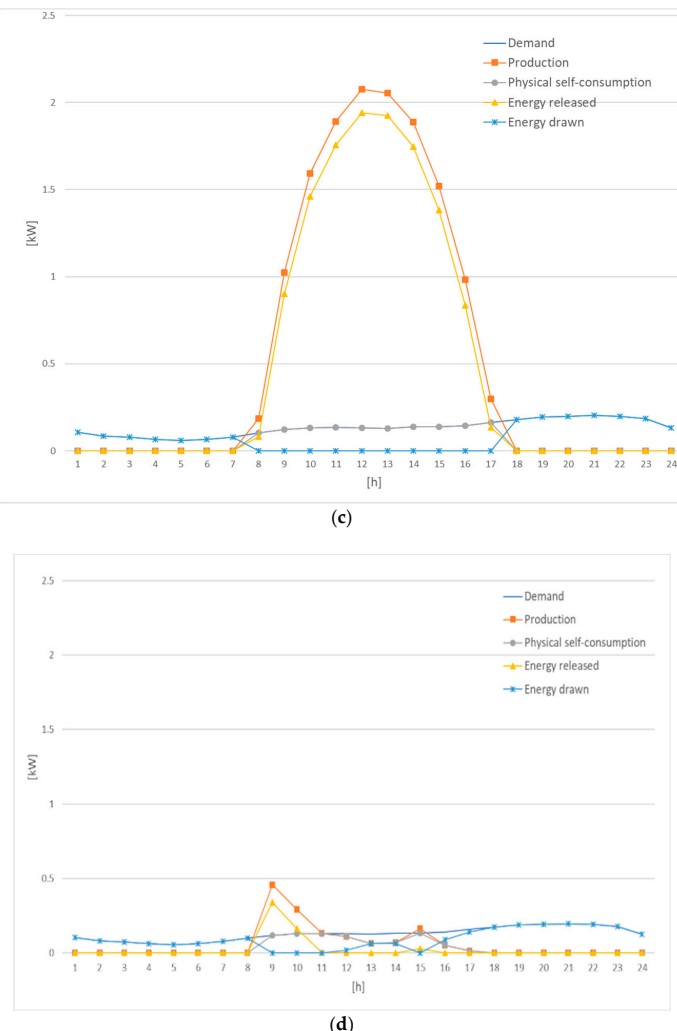

Figure 11. Electricity flows for building m1 during a typical (**a**) spring day (20 March 2022), (**b**) summer day (21 June 2022), (**c**) autumn day (23 September 2022), and (**d**) winter day (21 December 2022), for scenario #1.

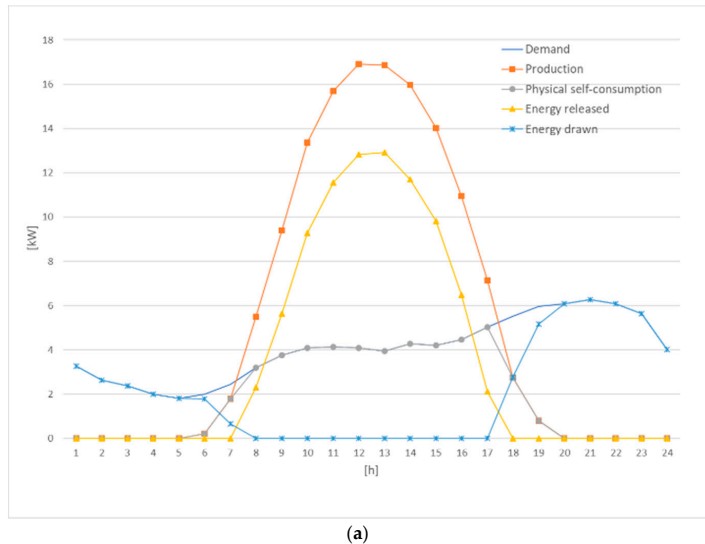

**Figure 12.** *Cont.*

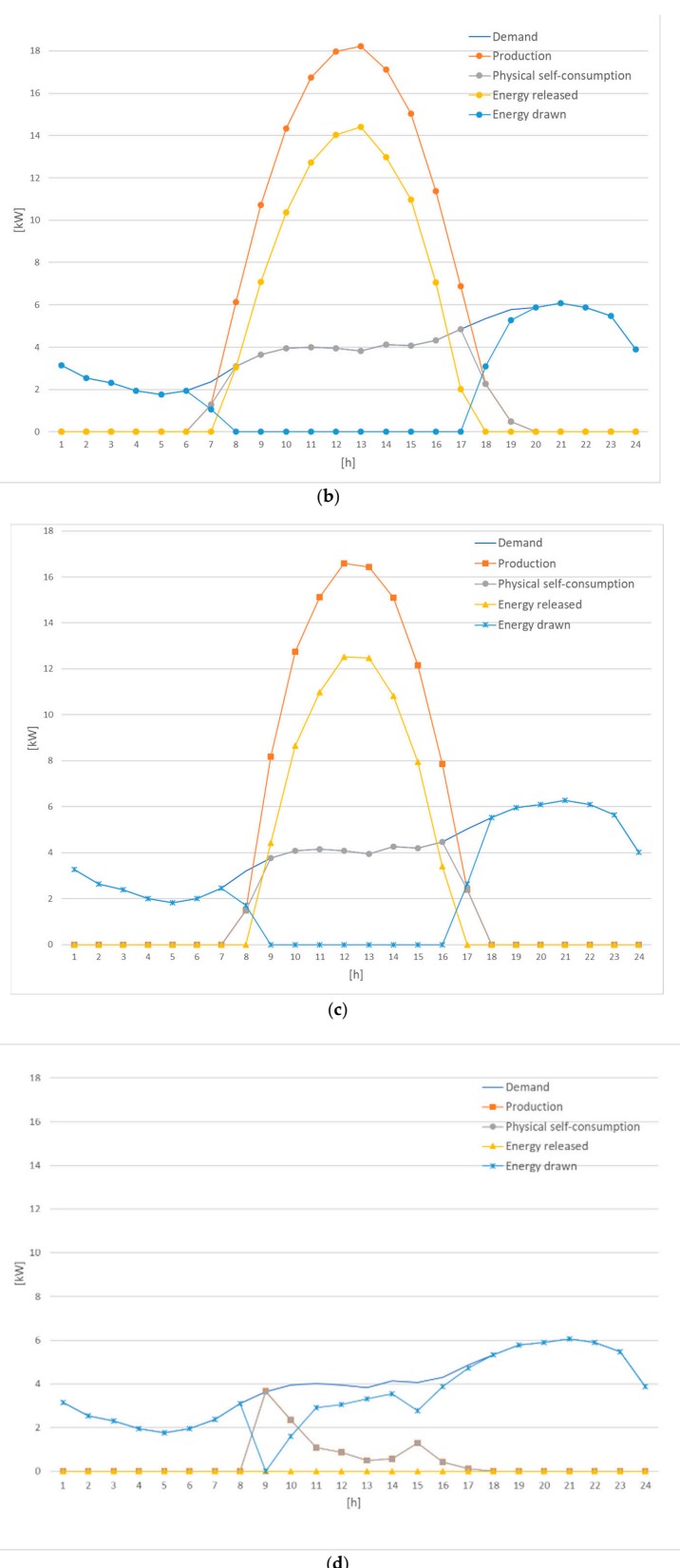

**Figure 12.** Electricity flows for building m15 during a typical (**a**) spring day (20 March 2022), (**b**) summer day (21 June 2022), (**c**) autumn day (23 September 2022), and (**d**) winter day (21 December 2022), for scenario #1.

As can be observed in Figure 11, for all the selected typical seasonal days, the production from PV is only partially used to cover the demand of the members themselves, and the higher contribution in terms of PV production is then released to the grid. This amount accounts for VSC, i.e., for the electricity sharing among members (and, therefore, for electricity that will be valorized from the GSE), and also for the electricity that will be sold to the power grid (and, therefore, simply paid in line with the unit price recognized from the electricity markets). These trends are common for all the four seasonal days, but with variations deriving from different PV production in summer, winter, and spring/autumn.

Figure 12 reports the same information shown in Figure 11, but for building m15, the REC member with the higher electricity demand.

Here, as can be noticed, the amount of electricity consumed under the PSC scheme is higher, due to the higher electricity demand of the building.

The optimized results for the scenarios #1.X, i.e., the scenarios in which there is variation of the unit cost of purchasing electricity from the grid (#1.1 and #1.2) or selling electricity to the power grid (#1.3 and #1.4) are reported in Table 5.

**Table 5.** Optimized results for scenarios #1.X.

|  | Scenario #1.1 | Scenario #1.2 | Scenario #1.3 | Scenario #1.4 |
|---|---|---|---|---|
| **NPV** | −€1,366,426.93 | −€358,387.55 | −€549,444.73 | −€1,175,369.75 |
|  | $\Delta$sc#1 = −58% | $\Delta$sc#1 = +58% | $\Delta$sc#1 = +36% | $\Delta$sc#1 = −36% |
| **Annual costs** | €169,393.46 | €101,637.38 | €135,515.42 | €135,515.42 |
| **Annual revenues** | €90,695.50 | €90,695.50 | €111,731.50 | €69,659.50 |

By comparing these scenarios with the base scenario #1, a change in the NPV is observed. In addition, this variation depends more significantly on the unit cost of purchasing electricity to the power grid (58%) than the unit price of selling electricity to the grid (a variation of 36%). Therefore, the purchase from the grid has a more evident impact on the NPV (negatively and positively).

## 5. Conclusions

This study presented a roadmap for the design, operation, and monitoring of RECs in Italy. The roadmap is inspired by the Deming Cycle, also known as Plan-Do-Check-Act, and consists of four main stages: Planning, Implementation, Monitoring, and Evaluation. The results of the case study in Sicily and the monitoring stage show how the design of electrical interactions among members of the RECs, as well as interactions between each member and the power grid, is essential to evaluate the revenues, establish to what extent the REC contributes to the reduction of carbon emissions, and determine their social impact calculated in terms of concrete assistance for families under energy poverty conditions. The results of the development phase I as part of the "do" stage reveal that the REC could contribute to a reduction of around 38% of the carbon emissions with 51 to 67 families helped through the revenues of the RECs, depending on the installed PV capacity. In addition, the electricity self-consumed under the PSC scheme and the electricity shared under the VSC scheme among members of the REC facilitate a sustainable transition, in which consumers substantially impact the electrical markets. Therefore, by following the roadmap, communities can contribute to a sustainable transition, reduce emissions, and help families in conditions of energy poverty.

Further improvements of the proposed research could be attained by including uncertainties related to the cost of electricity from the power grid and to the hourly demand of REC´s members. Secondly, while the model is built to cover the electrical demand, it does not consider the thermal consumption of buildings. In this direction, thermal sharing could represent a way to extend this study, for instance, via a district heating system. Future development could be devoted to evaluating the impact of new members in an already constituted and operating REC. Finally, implementing more comprehensive and tailored

control and monitoring procedures could positively affect the final stage that deals with continuous improvement of the REC.

**Author Contributions:** Conceptualization, A.F. and R.V.; Methodology, E.C., A.F. and R.V.; Software, E.C.; Formal analysis, R.V.; Investigation, E.C. and R.V.; Resources, E.C.; Data curation, E.C. and R.V.; Writing—original draft, E.C. and R.V.; Supervision, A.F. All authors have read and agreed to the published version of the manuscript.

**Funding:** This research has been financially supported by "Piano di incentivi per la ricerca di Ateneo 2020/2022 (Pia.ce.ri.)—Linea 2D"—University of Catania and by the SIS-RENEW research project (Piano di incentivi per la Ricerca 2020–2022).

**Data Availability Statement:** The data presented in this study are available on request from the corresponding author.

**Conflicts of Interest:** The authors declare no conflict of interest.

### Nomenclature

**Indices and sets**

| | |
|---|---|
| $m \epsilon M : \{m_0, m_1, \ldots, M\}$ | Member of the REC |
| $t \epsilon T : \{t_1, t_2, \ldots, T\}$ | Time-step |
| $y \epsilon Y : \{y_1, y_2, \ldots, Y\}$ | Year |
| **Subsets** | |
| $M' : \{m_1, \ldots, M\}$ | Distributed members |
| $T' : \{t_2, t_3, \ldots, T\}$ | Time-step for storage |
| **Parameters** | |
| $M$ | Number of REC members $[-]$ |
| $T$ | Number of time $-$ steps in a year $[-]$ |
| $Y$ | Lifespan of the investment $[-]$ |
| $En_{m,t}^{dem}$ | Electrical energy demand of member m at time $-$ step t$[kWh_e]$ |
| $conv_t$ | Electrical energy produced at time $-$ step t and normalized by the installed nominal power of the production technology $[kWh_e/kW_n]$ |
| $cap_m^{prod,min}$ | Minimum nominal capacity of a production technology physically connected to member m$[kW_n]$ |
| $cap_m^{prod,max}$ | Maximum nominal capacity of a storage physically connected to member m$[kW_n]$ |
| $cap_m^{stor,min}$ | Minimum nominal capacity of a storage physically connected to member m$[kW_n]$ |
| $cap_m^{stor,max}$ | Maximum nominal capacity of the storage physically connected to member m$[kW_n]$ |
| $Ecr$ | Energy to capacity ratio of the storage $[h]$ |
| $\eta^{charg}$ | Charging efficiency of the storage $[-]$ |
| $\eta^{discharg}$ | Discharging efficiency of the storage $[-]$ |
| $DoC$ | Depth of charge of the storage $[-]$ |
| $DoD$ | Depth of discharge of the storage $[-]$ |
| $B$ | Plan of the next period of storage $[-]$ |
| $\eta^{loss}$ | Electricity loss of the storage during one time $-$ step expressed as a percentage of the state of charge $[-]$ |
| $I$ | Return rate of the investment $[-]$ |
| $capex^{prod}$ | Unit investment cos t for production technology $[€/kW_n]$ |
| $capex^{stor}$ | Unit investment cost for storage $[€/kW_n]$ |
| $opex^{prod}$ | Maintenance and operating cos ts of production technology expressed as a percentage of the nominal capacity $[-]$ |
| $opex^{stor}$ | Maintenance and operating cos ts of storage expressed as a percentage of the nominal capacity $[-]$ |
| $buy^{PG}$ | Unit cos t of buying electricity from the power grid $[€/kWh_e]$ |
| $sell^{PG}$ | Unit price of selling electricity to the power grid $[€/kWh_e]$ |
| $Val$ | Valorization of electricity shared under the virtual scheme $[€/kWh_e]$ |
| $Inc$ | Incentives for electricity shared under the virtual scheme $[€/kWh_e]$ |
| $Efg$ | Emission factor of the power grid $[kgCO_2/kWh_e]$ |
| $Large$ | Sufficiently large number for the linearization procedure $[-]$ |
| $AED$ | Annual energy demand $[kWh/y]$ |

**Continuous variables**

| | |
|---|---|
| $CAP_m^{prod}$ | Installed nominal capacity of a production technology physically connected to member $m[kW_n]$ |
| $En_{m,t}^{prod}$ | Electricity from a production technology physically connected to member m at time $-$ step $t[kWh_e]$ |
| $En_{m,t}^{PSC,inst}$ | Electricity that is physically self $-$ consumed by member m at time $-$ step $t[kWh_e]$ |
| $En_{m,t}^{released}$ | Electricity released by member m to the power grid at time $-$ step $t[kWh_e]$ |
| $En_{m,t}^{drawn}$ | Electricity drawn from the power grid by member m at time $-$ step $t[kWh_e]$ |
| $En_t^{REC,released}$ | Electricity released by the REC to the power grid at time $-$ step $t[kWh_e]$ |
| $En_t^{REC,drawn}$ | Electricity drawn from the power grid by the REC at time $-$ step $t[kWh_e]$ |
| $En_t^{VSC}$ | Electricity used for virtual self $-$ consumption at time $-$ step $t[kWh_e]$ |
| $En_t^{REC,imp}$ | Electricity imported from the power grid at time $-$ step $t[kWh_e]$ |
| $En_t^{REC,exp}$ | Electricity exported to the power grid at time $-$ step $t[kWh_e]$ |
| $CAP_m^{stor}$ | Installed nominal capacity of a storage physically connected to member $m[kW_n]$ |
| $En_{m,t}^{discharged}$ | Electricity discharged by a storage physically connected to member m at time $-$ step $t[kWh_e]$ |
| $En_{m,t}^{charged}$ | Electricity used to charge a storage physically connected to member m at time $-$ step $t[kWh_e]$ |
| $En_{m,t}^{PSC}$ | Electricity that is self $-$ consumed at time $-$ step t for a storage installed by a member $m[kWh_e]$ |
| $SOC_{m,t}$ | State of charge of a storage physically connected member m at time $-$ step $t[kWh_e]$ |
| NPV | Net Present Values of the REC [€] |
| INV | Initial capital investment of the REC [€] |
| $CF_y$ | Cash flow of the REC at year $y[€]$ |
| $SAV_y$ | Savings of the REC at year $y[€]$ |
| $REV_y$ | Revenues of the REC at year $y[€]$ |
| $MOC_y$ | Maintenance and operating costs of the REC at year $y[€]$ |

**Binary variables**

| | |
|---|---|
| $X_m^{prod}$ | 1 if a production technology is physically connected to member m, 0 otherwise $[-]$ |
| $X_m^{stor}$ | 1 if a storage is physically connected to member m, 0 otherwise $[-]$ |
| $Y_{m,t}^{prod}$, $Y_{m,t}^{dem}$ | Binary variable used to find the minimum between produced electricity and demand (linearization technique) $[-]$ |
| $Y_t^{released}$, $Y_t^{drawn}$ | Binary variable used to find the minimum between released electricity and drawn energy (linearization technique) $[-]$ |

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
