# Peer review of "A Roadmap for the Design, Operation and Monitoring of Renewable Energy Communities in Italy"

_sustainability, doi:10.3390/su15108118_

Round 1
Reviewer 1 Report
The paper has some good contributions in terms of novelty and contribution. I have the following comments as follows:
1. The abstract must indicate the major findings in the last part of it. It should be more scientifically explained, not any informal or casual writing.
2. What are the fundamental findings that this paper can be published in this reputed journal? You must explain it.
3. The introduction should be explained based on the research gap, contribution, and organization of the paper.
4. Add the literature review separately with keyword specific.
5. There are several important findings in the literature in this direction. Therefore, it is important to obtain the novel findings of this research. There must be a comparative study with the following articles (Controllable energy consumption in a sustainable smart manufacturing model considering superior service, flexible demand, and partial outsourcing; Maintaining energy efficiencies and reducing carbon emissions under a sustainable supply chain management; Increasing Growth of Renewable Energy: A State of Art; An efficient sustainable smart approach to biofuel production with emphasizing the environmental and energy aspects) to show the major contributions and findings.
6. Keywords should be perfect. The abstract should contain the details of the study and the findings in a very constructive way.
7. The introduction should be based on the exact research gap, and the literature review should be based on the specific keywords-based review; finally, make an author's contribution table to show the novelty and effectiveness of the study. Show all referenced papers in the table to show the contribution of this study.
8. Please write the significant findings in conclusions. Do not mention all assumptions which have been indicated within the model.
9. Conclusions should be updated with more findings, limitations, and future extensions.
10. The applicability of the model should be explained. A real case study is required to prove the applicability of the study.
Author Response
Reviewer 1
The paper has some good contributions in terms of novelty and contribution.
We thank the Reviewer for her/his time spent in reviewing our manuscript and for the suggestions useful to improve the quality of our research. We sincerely hope the Reviewer may feel satisfied with our answers to the raised issues and the changes to the manuscript.
Questions 1.1 and 6.1
I have the following comments as follows:
- The abstract must indicate the major findings in the last part of it. It should be more scientifically explained, not any informal or casual writing.
- Keywords should be perfect. The abstract should contain the details of the study and the findings in a very constructive way.
Answer
We thank the Reviewer for her/his observation. As correctly suggested by the Reviewer, we have rewritten part of the abstract, as well as changed some of the keywords. We hope that the Reviewer may feel satisfied with the changes. We report here the revised abstract:
“Renewable energy communities (RECs) aim at achieving economic, environmental and social benefits for members and for the society. This paper presents a roadmap for the design, operation and monitoring of renewable energy communities in Italy, fundamental to guide and orient any stakeholder involved into the decision-making process of a REC. The roadmap is inspired by the Deming Cycle, also known as Plan-Do-Check-Act, which provides a framework for continuous improvement and standardization of the procedures. To demonstrate the practical application of the roadmap, a real case study is presented for Italian energy communities, making full adoption of data deriving from official database and using a real urban district as case study. Results confirm the validity of the roadmap in contributing to an aware design and implementation of RECs and orient political and energy decisions.”
Question 2.1
- What are the fundamental findings that this paper can be published in this reputed journal? You must explain it.
Answer
We thank the Reviewer for her/his comment and we tried to respond to the question here and in the revised version of our manuscript. Indeed, differently from the previously cited literature, this research provides guidance to these stakeholders during the all stages of the REC constitution process, by-passing the bias on the design and operational phases, usually the most studied within the scientific community. The whole process of a REC constitution depends on several factors, starting from normative and regulatory issues (with deriving red-tapes, especially in Italy) to technological and operational aspects. In addition, the impact of stakeholders’ decisions should be better regulated within the entire implementation process. As an example, municipalities may promote the inclusion of public buildings, as well as encourage the participation of residents and small enterprises, or ensure support to families in energy poverty conditions. There is therefore the need to organize these steps into a renewable energy community roadmap, i.e. into a structured and planned strategy, clarifying goals and objectives of RECs. The definition of a roadmap is beneficial for all members of the forthcoming REC, being it able not only to visualize the steps around its constitution, but also to visualize the temporal occurrence of any step as well as practical duties and expected outcomes. In addition, the roadmap fosters the communication among REC members, helping the identification of difficulties and trade-off decisions, as well as evaluating REC actual performances in terms of deviation from the targeted objectives, here declined in environmental, economic and social targets. As a further detail, and a novelty in the field of RECs, the presented roadmap lies its foundation in the so-calling Deming Cycle, known also as Plan-Do-Check-Act, notably diffused to elaborate strategies in the product enterprises. We have enlarged this discussion also in the manuscript and we sincerely hope the Reviewer may feel satisfied with our improvement.
Questions 3.1, 4.1 and 7.1
- The introduction should be explained based on the research gap, contribution, and organization of the paper.
- Add the literature review separately with keyword specific.
- The introduction should be based on the exact research gap, and the literature review should be based on the specific keywords-based review; finally, make an author's contribution table to show the novelty and effectiveness of the study. Show all referenced papers in the table to show the contribution of this study.
Answer
We thank the Review for these suggestions that help us in improving the quality of the manuscript. We have added a description of the structure of the paper at the end of the Introduction. Furthermore, we have expanded the literature review and drawn up a comparative study, shown in in Table 1, in order to highlight the main novelties of ours research. In this Table, we tried to give emphasis to some keywords to enhance the comparison and improve the readability of the paper. Then, a further paragraph has been introduced to reinforce the gap from the literature. We kindly ask the Reviewer to check the corrections within the manuscript and we hope to have satisfactorily responded to the requests,
Question 5.1
- There are several important findings in the literature in this direction. Therefore, it is important to obtain the novel findings of this research. There must be a comparative study with the following articles (Controllable energy consumption in a sustainable smart manufacturing model considering superior service, flexible demand, and partial outsourcing; Maintaining energy efficiencies and reducing carbon emissions under a sustainable supply chain management; Increasing Growth of Renewable Energy: A State of Art; An efficient sustainable smart approach to biofuel production with emphasizing the environmental and energy aspects) to show the major contributions and findings.
Answer
We thank the Reviewer for her/his attention. As requested, we inserted the papers and commented them in light of our research. The comparative table's layout (Table 1) was inspired by the ones in the articles provided by the Reviewer.
Questions 8.1 and 9.1
- Please write the significant findings in conclusions. Do not mention all assumptions which have been indicated within the model.
- Conclusions should be updated with more findings, limitations, and future extensions.
Answer
We thank the Reviewer for her/his suggestions. As requested, we removed the assumptions already indicated in the model and improved the discussion in terms of limitations and future extensions. We hope the Reviewer may positively judge the revision.
Question 10.1
- The applicability of the model should be explained. A real case study is required to prove the applicability of the study.
Answer
We thank the Reviewer for having highlighted this aspect. The case study is indeed derived from a real neighborhood in Catania. We totally agreed that it should be more detailed and specified. We kindly ask the Reviewer to check the modifications in the manuscript.

Reviewer 2 Report
1) Please polish the whole paper. Try to use short and clear expressions, not to be wordy.
2) In the literature section, I think it is more interesting to draw up a comparative table of the previous works in order to highlight the main features of the current work.
3) I suggest the authors to improve the introduction section. Authors should better highlight the objective of their work and to what extent it contributes to close a gap in the existing literature and/or practice.
4) Which is the innovative value of the contribution proposed by the authors?
5) The abstract is loosely written. It is not as informative as expected. A standard abstract must present, without leaving any doubt, the objective of the paper precisely; source of data (which is not present in your abstract) and analytical approach used; key findings and any policy implication and recommendations.
1) Please polish the whole paper. Try to use short and clear expressions, not to be wordy.
2) In the literature section, I think it is more interesting to draw up a comparative table of the previous works in order to highlight the main features of the current work.
3) I suggest the authors to improve the introduction section. Authors should better highlight the objective of their work and to what extent it contributes to close a gap in the existing literature and/or practice.
4) Which is the innovative value of the contribution proposed by the authors?
5) The abstract is loosely written. It is not as informative as expected. A standard abstract must present, without leaving any doubt, the objective of the paper precisely; source of data (which is not present in your abstract) and analytical approach used; key findings and any policy implication and recommendations.
Author Response
Reviewer 2
We thank the Reviewer for her/his time spent in reviewing our manuscript and for the suggestions useful to improve the quality of our research. We sincerely hope the Reviewer may feel satisfied with our answers to the raised issues and the changes to the manuscript.
Question 1.2
1) Please polish the whole paper. Try to use short and clear expressions, not to be wordy.
Answer
We thank the Reviewer for her/his comment. Accordingly, we have rephrased the sentences that were too long, favoring short and more communicate expressions. We kindly ask the Reviewer to check in the revised version of the manuscript.
Question 2.2
2) In the literature section, I think it is more interesting to draw up a comparative table of the previous works in order to highlight the main features of the current work.
Answer
We thank the Reviewer for this suggestion. Actually, we have added the Table in the Introduction and enlarged the discussion according to the Table itself. We kindly ask the Reviewer to check the modifications as well as the structure of the Table in the revised version of our manuscript.
Question 3.2
3) I suggest the authors to improve the introduction section. Authors should better highlight the objective of their work and to what extent it contributes to close a gap in the existing literature and/or practice.
Answer
We thank the Reviewer for her/his comment. As requested, we have reinforced the novelty of the manuscript by adding new contributions from the literature and posing more attention to the practical application of our approach. We kindly ask the Reviewer to check the Introduction section. We hope the Reviewer may feel satisfied with our revision.
Question 4.2
4) Which is the innovative value of the contribution proposed by the authors?
Answer
We thank the Reviewer for this comment, which forces to a better clarification of the main novelties and contribution with respect to the already existing literature in the field of renewable energy communities. Actually, in light of our bibliographic search, and thanks to discussion with stakeholders involved into the decision-making process of REC, we noticed a lack of well-defined steps, organized to help, orient and guide any future member of the REC. Therefore, we have elaborated a roadmap, which takes inspiration from the PDCA approach, also know as Deming Cycle, i.e. steps aiming at planning the activities, put the decisions into force, control how processes are evolving and eventually implement correction actions. We have added this description also in the Introduction and have also included Table 1, helpful to have an immediate understanding of what has been done from other Authors and how we aim at contributing. We kindly ask the Reviewer to check all this modification into the revised version of our manuscript, and we hope to have satisfactorily responded to the questions correctly raised by the Reviewer.
Question 5.2
5) The abstract is loosely written. It is not as informative as expected. A standard abstract must present, without leaving any doubt, the objective of the paper precisely; source of data (which is not present in your abstract) and analytical approach used; key findings and any policy implication and recommendations.
Answer
We thank the Reviewer for her/his suggestion. Accordingly, we have improved our abstract following the suggestions of the Reviewer. We report here the abstract, that we hope now respect the indications of the Reviewer.
“Renewable energy communities (RECs) aim at achieving economic, environmental and social benefits for members and for the society. This paper presents a roadmap for the design, operation and monitoring of renewable energy communities in Italy, fundamental to guide and orient any stakeholder involved into the decision-making process of a REC. The roadmap is inspired by the Deming Cycle, also known as Plan-Do-Check-Act, which provides a framework for continuous improvement and standardization of the procedures. To demonstrate the practical application of the roadmap, a real case study is presented for Italian energy communities, making full adoption of data deriving from official database and using a real urban district as case study. Results confirm the validity of the roadmap in contributing to an aware design and implementation of RECs and orient political and energy decisions.”

Reviewer 3 Report
Comments on “A roadmap for the design, operation and monitoring of renewable energy communities in Italy”.
Dear Authors,
The paper must be significantly improved. Please consider the following remarks:
Major comments:
(1) Please improve abstract part. Please answer the questions:
a) What problem did you study and why is it important?
b) What methods did you use?
c) What were your main results?
d) What conclusions can you draw from your results?
Please make your abstract with more specific and quantitative results while it suits broader audiences.
(2) Table 1. Please add source, year of data. Please explain or add abbreviation for “Max. PV installation [kWn] Max. battery capacity [kWh] Annual electricity demand [kWh]”
(3) Line 435 maximization NPV but in line 260-261 minimization NPV Please explain.
Minor comments:
(1) Figure 1. Please add source. Please explain relation month – year.
(2) Line 202. “The time-frame for participation can be arbitrarily set, but 3-4 weeks are recommended.” Please add source/reference for 3-4 weeks.
(3) Line 399-401: Please explain “The maximum installable PV capacity, reported in third column of Table 1, has been calculated by applying a typical conversion factor of 10 m2/kWn for tilted PV panels installed on available facing-south rooftop area in Sicily, as in [38].”
(4) figure 3, 4. How were data obtained? Please add equation, method, year. Please merge Figure 4 and 5: right and left scale
(5) Figure 6. Production -> specific yield ? Please add specific date
(6) Figure 8, 9, 10 Please add year for example 2022
(7) Figure 12, 13 Please add specific date
(8) Please improve references part in line with journal template.
Author Response
Reviewer 3
Comments on “A roadmap for the design, operation and monitoring of renewable energy communities in Italy”. Dear Authors, the paper must be significantly improved. Please consider the following remarks.
We thank the Reviewer for her/his time spent in reviewing our manuscript and for the suggestions useful to improve the quality of our research. We sincerely hope the Reviewer may feel satisfied with our answers to the raised issues and the changes to the manuscript.
Major comments:
Question 1.3
(1) Please improve abstract part. Please answer the questions:
a) What problem did you study and why is it important?
b) What methods did you use?
c) What were your main results?
d) What conclusions can you draw from your results?
Please make your abstract with more specific and quantitative results while it suits broader audiences.
Answer
We thank the Reviewer for pointing out the need to review the abstract. We tried to reformulate it, following these questions as possible guidelines for its structure. We now hope that the abstract respect the indications of the Reviewer. We report here the abstract, which was entirely rewritten.
“Renewable energy communities (RECs) aim at achieving economic, environmental and social benefits for members and for the society. This paper presents a roadmap for the design, operation and monitoring of renewable energy communities in Italy, fundamental to guide and orient any stakeholder involved into the decision-making process of a REC. The roadmap is inspired by the Deming Cycle, also known as Plan-Do-Check-Act, which provides a framework for continuous improvement and standardization of the procedures. To demonstrate the practical application of the roadmap, a real case study is presented for Italian energy communities, making full adoption of data deriving from official database and using a real urban district as case study. Results confirm the validity of the roadmap in contributing to an aware design and implementation of RECs and orient political and energy decisions.”
Question 2.3
(2) Table 1. Please add source, year of data. Please explain or add abbreviation for “Max. PV installation [kWn] Max. battery capacity [kWh] Annual electricity demand [kWh]”
Answer
We have added the source and the year of data, respectively in the text and in the Table, now Table 2. Furthermore, we have changed all abbreviations with full words while explaining more in depth their meaning.
Question 3.3
(3) Line 435 maximization NPV but in line 260-261 minimization NPV. Please explain.
Answer
We apologize for the typing mistake, now fixed by changing “minimization” with “maximization” in the formulation of the objective function.
Minor comments:
Question 1.3
(1) Figure 1. Please add source. Please explain relation month – year.
Answer
We do not have a source for this Figure, since it has been elaborated and created by us. Now we have added this comment in the Figure description, as well as explained the relation month-year.
Question 2.3
(2) Line 202. “The time-frame for participation can be arbitrarily set, but 3-4 weeks are recommended.” Please add source/reference for 3-4 weeks.
Answer
We thank the Reviewer for this comment. Actually, this timeframe does not derive from a source, rather from experiences of the Authors in this field, especially in the Italian context. We have specified this in the manuscript.
Question 3.3
(3) Line 399-401: Please explain “The maximum installable PV capacity, reported in third column of Table 1, has been calculated by applying a typical conversion factor of 10 m2/kWn for tilted PV panels installed on available facing-south rooftop area in Sicily, as in [38].”
Answer
We have re-phrased the quoted text with a clearer and more detailed description, as suggested by the Reviewer.
Question 4.3
(4) figure 3, 4. How were data obtained? Please add equation, method, year. Please merge Figure 4 and 5: right and left scale
Answer
We have added the source for the typical daily load curve in the description of Figure 3. The scaling technique used to obtain data plotted in Figure 3 and 4 is the same as the one applied in [37], that is cited in the paper. The Reviewer can refer to this source to find a detailed description of methods and equations, as we also detailed it in the manuscript. Finally, we have merged Figure 4 and 5 in the same graph with two different scales.
Question 5.3
(5) Figure 6. Production -> specific yield? Please add specific date
Answer
We changed the description of Figure 6. to make it clearer that it refers to the specific yield for 1 kWn of PV installed nominal capacity. We also added the specific dates for each typical day in the description, as suggested by the Reviewer.
Question 6.3
(6) Figure 8, 9, 10 Please add year for example 2022
Answer
We added the year in the descriptions of Figure 8, 9, 10 and 11, as suggested by the Reviewer.
Question 7.3
(7) Figure 12, 13 Please add specific date
Answer
We added the specific dates in the descriptions of Figure 12 and 13, as suggested by the Reviewer.
Question 8.3
(8) Please improve references part in line with journal template.
Answer
We improved the References, as suggested by the Reviewer.

Reviewer 4 Report

I suggest the authors use a proofreading service since minor English changes must be made.
Author Response
Reviewer 4
I suggest the authors use a proofreading service since minor English changes must be made.
File attached.
We thank the Reviewer for her/his time spent in reviewing our manuscript and for the suggestions useful to improve the quality of our research. We sincerely hope the Reviewer may feel satisfied with our answers to the raised issues and the changes to the manuscript.
Answer
We have corrected flaws and improved Grammar and English flow. We sincerely hope that the Reviewer may feel satisfied with the implemented corrections.
Regarding the comments in the attached file:
Article: A roadmap for the design, operation and monitoring of renewable energy communities in Italy. In this paper, the authors present a roadmap for the design, operation and monitoring of renewable energy communities in Italy.
Abstract
The abstract is structured according to the Instructions of the journal.
Answer
We thank the Reviewer for her/his comment.
Introduction
The Introduction section needs to be improved. The authors should highlight the main conclusions and contributions of the research at the end of this section. Furthermore, the authors can add a final paragraph for summarising the content of the next sections.
Answer
We thank the Reviewer for her/his suggestion. We have improved the Introduction, starting with the flow of the discussion to enhance the readability and also adding a complementary Table to highlight the novelty of our manuscript. In addition, as suggested by the Reviewer, we have also added a paragraph with a summary of the content of the manuscript.
Material and Methods
The authors well describe the Materials and Methods section.
Answer
We thank the Reviewer for her/his positive comment.
Results
The results are highlighted and discussed in detail.
Answer
We thank the Reviewer for her/his positive comment.
Conclusions
In general, the objective, the main results and discussions, contributions, and limitations should be mentioned in this section. However, this paper does not present the contributions and limitations of the study, and only summarizes the results in the Conclusion. Therefore, it is recommended that the author strengthen it, pointing out the scientific contributions and limitations of the study. Furthermore, it is essential that the authors point out the research opportunities that may have arisen from this study.
Answers
We thank the Reviewer for her/his comment. Accordingly, we have now revised the manuscript, adding limitations and future research as well as the main implications of our study. We sincerely hope the Reviewer may appreciate the changes.
References
References cited – Please, correctly cite all citations in the text when the author's name is cited in the text, for example in line 90:
Stentati et al. proposed an optimization model to compute the battery charging/discharging polices while selling points of flexible loads and controllable generators for a REC in the Italian incentive systems [13].
Change to:
Stentati et al. [13] proposed an optimization model to compute the battery charging/discharging polices while selling points of flexible loads and controllable generators for a REC in the Italian incentive systems.
Answers
We thank the Reviewer for her/his comment. Accordingly, we improved the citation style throughout the manuscript.

Reviewer 5 Report
The proposed article explores an important topic. The authors propose a roadmap for the design, operation and monitoring of renewable energy communities in Italy. An interesting study has been done. The topic is also relevant for the readers of the magazine. To continue the review, it is necessary to conduct an audit. I see the following issues:
1. Introduction should be rewritten.
2. It must contain the purpose of the study, scientific novelty and hypothesis.
3. The authors must specify the rationale for the hypothesis.
4. Before the "Conclusion" add a separate paragraph "Discussion".
5. Language must be checked.
Summary.
In my opinion, the article can be published after completion.
Author Response
Reviewer 5
The proposed article explores an important topic. The authors propose a roadmap for the design, operation and monitoring of renewable energy communities in Italy. An interesting study has been done. The topic is also relevant for the readers of the magazine. To continue the review, it is necessary to conduct an audit.
We thank the Reviewer for her/his time spent in reviewing our manuscript and for the suggestions useful to improve the quality of our research. We sincerely hope the Reviewer may feel satisfied with our answers to the raised issues and the changes to the manuscript.
Questions
I see the following issues:
- Introduction should be rewritten.
- It must contain the purpose of the study, scientific novelty and hypothesis.
- The authors must specify the rationale for the hypothesis.
- Before the "Conclusion" add a separate paragraph "Discussion".
- Language must be checked.
Summary.
In my opinion, the article can be published after completion.
Answers
We kindly thank the Reviewer for her/his comments and observations. Accordingly, we have implemented all the requested modifications. In particular:
1/2. The Introduction has been improved and, in particular, a dedicated table summarizing the main contributions from the literature has been added to enhance the readability of the manuscript and reinforce the novelty of our proposed roadmap, especially in the field of energy communities.
- More attention, as correctly pointed out by the Reviewer, has been posed to the hypotheses and rationale behind the model, for which we kindly ask the Reviewer to check the modifications in the revised version of our manuscript.
- As a unique point, we maintained the “Results and discussion” Section, since we prefer to highlight specific considerations behind each different result, however to reinforce the discussion we have added new considerations in the Conclusions, specifically with respect to limitations and future applications.
- Language and style have been improved, also checking grammar and flow.
We again thank the Reviewer for her/his comments and we sincerely hope she/he may feel satisfied with our revision.

Round 2
Reviewer 1 Report
The paper can be accepted for publication.
Author Response
We thank the Reviewer for her/his positive evaluation.
Reviewer 3 Report
Reference part should be improved in line with journal template.
Figure 5. Please add information "where?"
Line 428 typo "]"
Abstract part should be improved. Please add the main results.
Introduction part: Perhaps in this context it is worth quoting the results of research from other countries, which are discussed, for example, in works: DOI 10.3390/en14020319 DOI 10.3390/en14113226 and https://doi.org/10.3390/en14040931
Equation no. 41. Please use symbol instead of exact number "2700".
Author Response
Reviewer 3
Reference part should be improved in line with journal template.
Answer: Although the journal template has already been used, we will ask the editor of the journal for major clarifications. We hope the Reviewer won't consider it to be a major issue.
Figure 5. Please add information "where?"
Answer: We have added the district’s location both in the text and in the label of Figure 5., as suggested by the Reviewer.
Line 428 typo "]"
Answer: We have corrected the typo by replacing the square brackets in Eq. (4) with round ones.
Abstract part should be improved. Please add the main results.
Answer: We have improved the abstract part by adding the main results of the study, as suggested by the Reviewer.
Introduction part: Perhaps in this context it is worth quoting the results of research from other countries, which are discussed, for example, in works: DOI 10.3390/en14020319 DOI 10.3390/en14113226 and https://doi.org/10.3390/en14040931
Answer: We have quoted these three works at lines 110-113, 115-117 and line 121. We thank the Reviewer for the suggestion which has improved the quality of our literature review.
Equation no. 41. Please use symbol instead of exact number "2700".
Answer: We have replaced the exact number 2700 in Eq. (41) with the symbol AED (annual energy demand), which has also been added in the parameter table at the beginning of the paper.